# SpikF: Spiking Fourier Network for Efficient Long-term Prediction

**Wenjie Wu** [1]  **Dexuan Huo** [1]  **Hong Chen** [1]

## Abstract

Spiking Neural Networks (SNNs) have demonstrated remarkable potential across many domains, including computer vision and natural language processing, owing to their energy efficiency and biological plausibility. However, their application in long-term prediction tasks remains underexplored, which is primarily due to two critical challenges: (1) current SNN encoding methods are unable to effectively encode long temporal information, leading to increased computational complexity and energy consumption; (2) though Transformer-based models have achieved state-of-the-art accuracy in temporal prediction tasks, the absence of proper positional encoding for spiking self-attention restricts Spiking Transformer from effectively utilizing positional information, resulting in performance degradation. To address these challenges, we introduce an attention-free framework, **Spik**ing **F**ourier Network (**SpikF**), that encodes input sequences in patches and employs an innovative frequency domain selection mechanism to effectively utilize the sequential properties of time-series data. Extensive evaluations on eight well-established long-term prediction datasets demonstrate that SpikF achieves an averaged $1.9\%$ reduction in Mean Absolute Error (MAE) compared to state-of-the-art models, while lowering total energy consumption by $3.16\times$. Our code is available at https://github.com/WWJ-creator/SpikF.

## 1. Introduction

Spiking Neural Networks (SNNs), as the third generation of neural networks (Maass, 1997), emulate the brain's information processing mechanisms through discrete spike events rather than continuous value computations (Roy et al., 2019).

This biologically-inspired approach addresses the critical limitation of Artificial Neural Networks (ANNs) in energy efficiency. While ANNs have dominated machine learning applications, their substantial computational demands create challenges for resource-constrained environments. By leveraging event-driven processing where computations occur only during sparse spike events, SNNs achieve comparable performance to ANNs with significantly reduced energy consumption (Roy et al., 2019). This efficiency benefit makes SNNs particularly suitable for edge computing and real-time processing applications (Maass, 1996).

With these advantages, SNNs have made significant strides across various machine learning domains, particularly in computer vision. In image classification tasks, SNNs integrated with Transformer architectures (Vaswani et al., 2017; Zhou et al., 2023; 2024) have achieved performance on par with conventional ANNs. Similarly, in object detection tasks, notable works such as Spiking-YOLO (Kim et al., 2020) and SpikeYOLO (Luo et al., 2024) have further demonstrated the potential of SNNs. Beyond computer vision, SNNs have also shown promising results in natural language processing, exemplified by advancements like SpikeGPT (Zhu et al., 2024), highlighting their versatility across diverse domains.

However, the application of SNNs to long-sequence prediction remains relatively unexplored. Though recent work (Lv et al., 2024b), has investigated SNN performance in short-sequence prediction tasks, comprehensive benchmarking of SNNs on popular long-sequence prediction datasets (Electricity, Weather, ETT, Traffic and Exchange) has not been extensively established.

The limited exploration of SNNs in long-term prediction tasks is due to two critical challenges. Firstly, the current spike encoding mechanisms employed in temporal prediction tasks, such as delta encoding (Amir et al., 2017), convolution-based encoding (Lv et al., 2024b), and linear-based encoding (Lv et al., 2024b), treat the input sequence as a monolithic entity. These approaches increase the hidden dimension of the model, leading to a substantial computational burden that undermines the inherent energy efficiency advantages of SNNs. Secondly, while ANN Transformer-based models have achieved remarkable accuracy in long-term prediction tasks, such as Autoformer (Wu et al., 2022),

---

[1]Tsinghua University, Beijing, China. Correspondence to: Hong Chen <hongchen@tsinghua.edu.cn>.

*Proceedings of the 42nd International Conference on Machine Learning*, Vancouver, Canada. PMLR 267, 2025. Copyright 2025 by the author(s).

FEDformer (Zhou et al., 2022), PatchTST (Nie et al., 2023) and iTransformer (Liu et al., 2024), the traditional self-attention mechanism suffers from permutation-invariance (Zeng et al., 2023), as detailed in Appendix A.1. This limitation necessitates the incorporation of positional encoding (Vaswani et al., 2017) to effectively embed positional information into the input data. However, positional encoding in Spiking Transformers remains underexplored. Despite innovative approaches, such as central pattern generator-inspired spiking positional encoding (Lv et al., 2024a), Spikformer utilizing this mechanism exhibit only marginal performance improvements and remain unable to deliver satisfactory results in temporal prediction tasks. In fact, Spikformer equipped with central pattern generator-inspired positional encoding demonstrate suboptimal performance compared to simpler SNN architectures, such as SpikeTCN (Tavanaei & Maida, 2017) and SpikeRNN (Kim et al., 2019), incurring accuracy losses of 2.6% and 7.7% in Root Relative Squared Error (RSE), respectively. This performance gap underscores the current limitations of spiking positional encoding techniques in effectively embedding positional information, thereby hindering the potential of Spiking Transformers in long-term prediction tasks.

In order to address the challenges, this paper proposes **Spik**ing **F**ourier Network (**SpikF**), an attention-free architecture. Specifically, SpikF adapts Spiking Patch Encoding from Patch Embedding in PatchTST (Nie et al., 2023) and incorporates a Spiking Frequency Selection mechanism to model dependencies between patches, replacing the self-attention mechanism and naturally leveraging the sequential properties of time-series data. Our approach outperforms existing SOTA models by 1.9% across eight well-established long-term prediction real-world datasets while reducing energy consumption by 75.05% in a representative study on the ECL dataset. SpikF provides a computationally efficient framework for long-term forecasting, demonstrating exceptional suitability for energy-constrained scenarios and edge device deployment. Our key contributions include:

- We propose a Spiking Patch Encoder which divides input sequences into patches and independently encodes each sub-series into binary spike trains. By reducing computational complexity, this design enables SNNs to effectively handle longer input sequences, thereby enhancing their ability to utilize historical information.

- We propose a novel Spiking Frequency Selection mechanism that identifies and selects critical components from input sequences to enhance prediction performance. This mechanism establishes long-range temporal dependencies not only within the input sequence but also between the sequence and prediction targets, leveraging the positional information inherently embedded in the Fourier Transform.

- We conduct empirical evaluations across eight real-world long-term time-series benchmark datasets, demonstrating that SpikF achieves superior accuracy over SOTA ANN models, with a 1.9% reduction in MAE. Our extensive experiments across Electricity, Weather, ETT, Traffic and Exchange datasets validate the robustness of our approach and establish a foundational SNN benchmark for the research community. To our knowledge, SpikF is the first SNN-based benchmark providing full coverage across these datasets. It not only expands the scope of SNN research by establishing a benchmark for future SNN-based temporal prediction models, but also provides an efficient and innovative solution for the temporal prediction research community.

## 2. Related Works

### 2.1. Time-series Prediction

The time-series prediction methodologies have developed from traditional statistical approaches, such as ARIMA (Box et al., 1978) and exponential smoothing (Hyndman et al., 2008), to advanced deep learning architectures. These include Temporal Convolutional Networks (TCNs) (Bai et al., 2018; donghao & wang xue, 2024), Recurrent Neural Networks (RNNs) (Zhang et al., 2023; Jia et al., 2024), and Transformer-based models. In recent years, attention-based models (Vaswani et al., 2017) have emerged as advanced solutions for time-series forecasting. Significant progress in this field has been made with models such as Informer (Zhou et al., 2021) and Autoformer (Wu et al., 2022), which tackle the quadratic time complexity of self-attention and enhance temporal dependency modeling. More recent innovations, such as PatchTST (Nie et al., 2023), enhance the local semantic information of time-series by dividing input sequences into patches, while iTransformer (Liu et al., 2024) enhances the modeling of dynamic correlations among different variates through inverted token mechanisms. Despite their remarkable performance, these models often require substantial computational resources and energy, which restricts their applicability in resource-constrained environments.

In contrast, SNNs have garnered limited but growing attention for time-series prediction. Several exploratory studies have demonstrated their potential, including a two-phase SNN for electricity load prediction (Kulkarni et al., 2013), a polychronous spiking network for financial data prediction (Reid et al., 2014), a NeuCube-based (Kasabov, 2014) architecture for crop yield prediction (Bose et al., 2016), and a multi-modal SNN architecture for financial stock prediction (AbouHassan et al., 2023). However, none of these works addresses the challenge of long-term prediction. A recent study proposed an SNN framework tailored for short-

sequence prediction tasks (Lv et al., 2024b), representing a significant advancement in this emerging field. Despite these progresses, research on SNNs for long-term forecasting remains relatively limited, and a comprehensive benchmark for evaluating their performance on widely-used long-sequence datasets is still absent. This gap arises from the inherent challenges faced by SNNs, including the efficient encoding of long sequences and the inadequate utilization of sequential information.

## 2.2. Frequency-based Methods in Time-series Forecasting

Time-series forecasting has seen significant advancements through the integration of frequency domain analysis, which allows for more efficient processing and enhanced forecasting performance.

**FITS** (Xu et al., 2024) employs a complex linear layer to interpolate in the frequency domain, discarding high-frequency noise through a cutoff frequency strategy, thus achieving superior performance compared to DLinear.

**FreTS** (Yi et al., 2023) further expands the utilization of frequency domain features through frequency domain MLPs, which facilitate time-series forecasting by providing a global view and energy compaction.

**FEDformer** (Zhou et al., 2022) employs frequency domain selection to generate sparse attention, thereby reducing computational complexity and capturing detailed structures of time-series data.

**FilterNet** (Yi et al., 2024) adapts filters from the signal processing field, developing two kinds of filters to weaken or strengthen specific frequency components, thus benefiting from the utilization of the full spectrum.

Most previous works have focused on applying FFT to the entire series, which facilitates the utilization of high-frequency components. In contrast, our approach employs patch and grouping mechanisms to enhance the utilization of low and middle-frequency components in the original series, where local information is emphasized by the spiking patches, which are discussed in following sessions.

## 2.3. Spiking Neural Networks

SNNs have gained increasing attention due to their energy efficiency and biological plausibility (Roy et al., 2019). Unlike ANNs that process continuous activation values, SNNs operate using discrete spikes, closely mimicking the behavior of the human brain.

**Spiking Neuron Model** One of the most popular spiking neuron models is Leaky Integrate-and-Fire (LIF) Neuron (Abbott, 1999), due to its simplicity and widely utilization in recent research (Zhou et al., 2023; 2024; Lv et al., 2024b).

The LIF neuron receives the resultant current, integrates it to accumulate membrane potential, and compares this potential with a predefined threshold to determine whether to generate a spike. The membrane potential $V[t]$ evolves according to:

$$U[t] = V[t-1] + \frac{1}{\tau_m} \left( I[t] - V[t-1] + V_{rest} \right) \quad (1)$$

$$S[t] = H \left( U[t] - V_{th} \right) \quad (2)$$

$$V[t] = U[t] \left( 1 - S[t] \right) + V_{rest} S[t] \quad (3)$$

where $\tau_m$ is the membrane time constant, $V_{rest}$ is the resting potential, $H$ represents the Heaviside step function and $I[t]$ is the input current which can be calculated by:

$$I[t] = W S_{pre}[t] \quad (4)$$

where $S_{pre}[t]$ denotes the spikes emitted by the pre-synaptic neurons, and $W$ represents the synaptic weights between different layers.

**SNN Training Method** The indifferentiable nature of the Heaviside step function makes it challenging to train deep SNNs. There are two popular approaches to train deep SNNs. One is ANN2SNN conversion (Cao et al., 2015; Bu et al., 2021; Wang et al., 2022), which trains an ANN model and then converts the model into rate-coded SNN model. However, this approach usually needs multiple simulating time steps to achieve comparable accuracy with the original ANN model, which causes large latency (Han et al., 2020). The other approach utilizes surrogate gradient function to approximate the derivative of the Heaviside step function, and thus conducting spatio-temporal backpropagation (STBP) to update the network parameters. In our work, we adopt the latter approach.

**Fast Fourier Transform (FFT) in SNNs** While SNNs excel at processing temporal sequences, owing to their intrinsic membrane dynamics, their ability to capture global temporal features is inherently limited by two fundamental characteristics: the exponential decay factor of membrane potentials and the reset mechanism of neuronal activation, which is discussed in Appendix A.2. Recent advancements (Lopez-Randulfe et al., 2022; Orchard et al., 2021) have offered a promising solution to this limitation through the integration of Fast Fourier Transform (FFT) with SNN architectures.

Study (Lopez-Randulfe et al., 2022) has demonstrated that matrix multiplication can be represented by a spiking linear layer with an appropriately defined weight matrix. Accordingly, they initially express the FFT as a series of matrix multiplications and subsequently employ an SNN with an equivalent number of layers to avoid the challenges associated with floating-point operations.

Meanwhile, study (Orchard et al., 2021) leverages the membrane dynamics of the Resonate-and-Fire neuron, an extension of the LIF model, to naturally perform the Fourier Transform. This approach also successfully gets rid of the need for additional floating-point operations.

## 3. Methodology

### 3.1. Problem Setup

The multivariate long-term forecasting framework can be formulated as follows: given an input time-series $x^{1:L} \in \mathbb{R}^{L \times D}$ of length $L$ with $D$ dimensions, our object is to predict the subsequent $H$ values, denoted as $y^{1:H} \in \mathbb{R}^{H \times D}$.

### 3.2. Overview

The overall architecture of SpikF is illustrated in Figure 1. SpikF comprises three components: (1) a Spiking Patch Encoder (SPE) for transforming continuous-valued input time-series into binary spike representations; (2) a feature extraction module with a Spiking Frequency Selector (SFS) to select important frequency components and convolutional layers to enhance local semantic information; (3) an MLP Decoder to generate the final output.

**Spiking Patch Encoder**  Although convolutional and delta encoding mechanisms have been proposed by (Lv et al., 2024b), they exhibit limitations in encoding extended historical information. To solve the problem, we introduce a novel patch-based encoding mechanism that processes input sequences in segmented patches, thereby facilitating a computationally efficient representation of temporal patterns.

**Spiking Frequency Selector**  As we know, self-attention mechanisms have gained widespread adoption across various machine learning domains, but their inherent permutation-invariance renders them inadequate in capturing positional information in time-series data. FFT-based models (Zhou et al., 2022; Xu et al., 2024) are effective at capturing sequential properties, but their frequency modes are restricted to maintain computational efficiency. Specifically, FEDformer (Zhou et al., 2022) relies on fixed or randomly selected frequency modes, while FITS (Xu et al., 2024) adopts a cut-off frequency strategy to discard high-frequency components. Both of the two approaches employ artificial selection strategies, failing to take the characteristics of input sequences into consideration. To address these problems, we propose a novel frequency domain selection mechanism that dynamically identifies relevant frequency components of spike trains by leveraging the binary nature of spikes. This approach not only enhances the utilization of sequential properties but also alleviates the burden on researchers to manually determine key frequencies for the input sequence.

### 3.3. Spiking Patch Encoder

Our spike encoding pipeline comprises three key stages. First, the input time-series signal is segmented into patches, each of which is processed through a shared learnable linear layer that maps the data into a higher-dimensional space, as being put forward by PatchTST (Nie et al., 2023) to enhance the better utilization of local semantic information. These transformed patches are then temporally upsampled by a factor of $T_s$ using a zero-order hold (ZOH) strategy, increasing their temporal resolution. The upsampled signals serve as input currents to the LIF neurons and are transformed into binary spike trains, converting the analog information into neuromorphic representations.

The $x^{1:L}$ will be divided into patches:

$$p^k = x^{\frac{L}{N}(k-1)+1:\frac{L}{N}k} \tag{5}$$

where $N$ is the number of patches and we assume $L$ can be divided by $N$.

Then each patch will be processed by a spiking linear layer:

$$S_{enc}^{T_s(k-1)+1:T_s k} = \mathcal{SN}(\text{BN}(\text{LN}(p^k))) \tag{6}$$

where LN denotes the linear transformation layer, BN represents batch normalization, and $\mathcal{SN}$ denotes the spiking neuron layer.

### 3.4. Spiking Frequency Selector

The encoded spike trains are processed through two blocks in parallel to enhance modeling of positional information and energy efficiency. Initially, the spiking patches are grouped at fixed intervals. In the first block, the grouped patches are transformed into frequency components using a Spiking Fast Fourier Transform (S-FFT) layer. In the second block, the grouped patches are converted into frequency-domain spiking selector via a spiking selector generator, which consists of a linear projection layer, a batch normalization layer, and a spiking neuron layer. To improve the transmission efficiency of the spiking selector, a spiking max-pooling layer is utilized. The frequency spectrum is then selected by the spiking selector. The selected frequency components are reconstructed in the time domain using a grouped Spiking Inverse Fast Fourier Transform (S-iFFT) layer. The resulting real-valued time-series are the input to the LIF neurons, generating spike trains with reduced noise. The sparsity of spike trains contributes to computational efficiency in both S-FFT and inverse operations. Additionally, the S-FFT enables the spiking selector to maintain a global receptive field across the entire time-series while leveraging positional information.

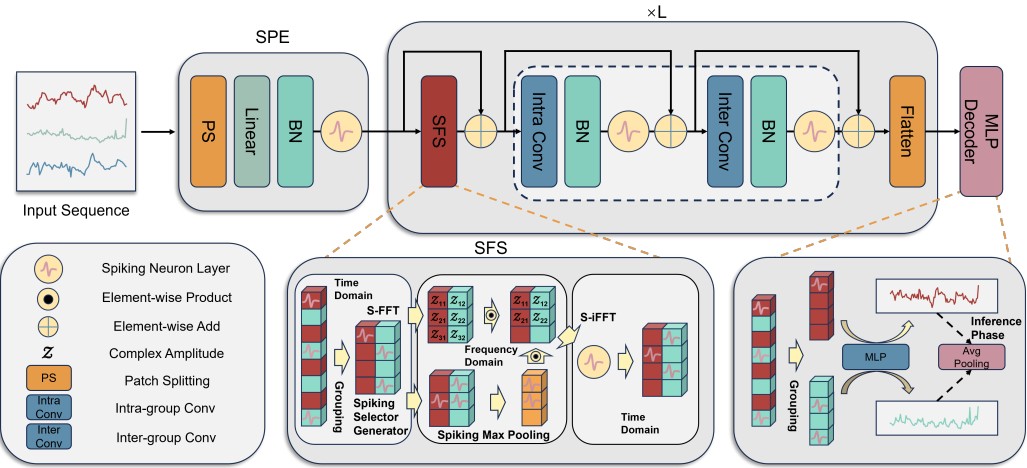

*Figure 1.* Architecture overview of the proposed SpikF framework. (a) The Spiking Patch Encoder (SPE) converts continuous time-series data into discrete spike train representations through patch-based processing. (b) The Spiking Frequency Selector (SFS) integrates S-FFT operations across spike train patches, thereby facilitating the natural utilization of positional information. (c) The decoder reconstructs real-valued predictions from the processed spike patterns by employing an MLP architecture.

The spiking patches generated by the Spiking Patch Encoder are first grouped at fixed intervals:

$$\mathbf{G}^i = \{S_{enc}^i, S_{enc}^{i+g}, \ldots, S_{enc}^{i+(\frac{NT_s}{g}-1)g}\} \tag{7}$$

where $\mathbf{G}^i$ denotes $i$th group of patches and $g$ represents the number of groups.

Then each $\mathbf{G}^i$ is transformed into the frequency domain through the S-FFT algorithm:

$$\mathbf{F}^i = \mathcal{SF}_g(\mathbf{G}^i) \tag{8}$$

where $\mathcal{SF}_g$ denotes the grouped S-FFT operation.

In the meantime, $\mathbf{G}^i$ passes through a spiking linear layer to generate frequency domain spikes:

$$\mathcal{M}_{sel}^i = \mathcal{SN}(\mathrm{BN}(\mathrm{LN}(\mathbf{G}^i))) \tag{9}$$

Then, a spiking max pooling layer is used to emphasize the mutual key frequencies of different groups:

$$\mathbf{M}_{sel} = \mathrm{SMP}(\mathcal{M}_{sel}^1, \mathcal{M}_{sel}^2, \ldots, \mathcal{M}_{sel}^g) \tag{10}$$

where $\mathrm{SMP}$ represents the Spiking Max-pooling layer.

Finally, $\mathbf{M}_{sel}$ selects the important frequency components through the Hadamard product, and these components are sent back to the time domain through S-iFFT:

$$\mathbf{H}_{\mathcal{F}}^i = \mathcal{SF}_g^{-1}(\mathbf{M}_{sel} \odot \mathbf{F}^i) \tag{11}$$

where $\mathcal{SF}_g^{-1}$ denotes the grouped S-iFFT operation and $\odot$ denotes the Hadamard element-wise product.

Then $\mathbf{H}_{\mathcal{F}}^i$ serves as the stimulation current to spiking neurons, generating spikes with reduced noise:

$$\mathbf{S}_{\mathcal{F}}^i = \mathcal{SN}(\mathrm{BN}(\mathbf{H}_{\mathcal{F}}^i)) \tag{12}$$

### 3.5. Multilayer Perceptron (MLP) Decoder

As is well known, it is challenging to realize high temporal resolution of spike trains for the spike decoder, as directly applying an MLP with a full receptive field to reconstruct continuous-valued outputs would increase both the number of learnable parameters and computational overhead. To overcome this challenge while maintaining the temporal integrity of the spiking patches, we introduce a temporal synchronization mechanism.

We first synchronize the temporal resolution between spiking patches and prediction series through upsampling the prediction target. Then, both spikes and prediction targets are divided into $T_s$ groups, and the spike trains are processed through an MLP to reconstruct the continuous-valued prediction. During the training phase, the loss function is defined as:

$$\mathcal{L} = \frac{1}{T_s} \sum_{k=1}^{T_s} \left\| \mathrm{MLP}(\mathbf{S}^k) - \mathbf{Y} \right\| \tag{13}$$

where $\mathbf{S}^k$ represents the spike train in the $k$th group and $\mathbf{Y}$ denotes the target prediction series.

During the inference phase, the predicted sequence undergoes dimensionality reduction through average pooling, and

the final prediction sequence is generated. This process is formally expressed as:

$$\hat{\mathbf{Y}} = \frac{1}{T_s} \left( \sum_{k=1}^{T_s} \mathrm{MLP}(\mathbf{S}^k) \right) \tag{14}$$

where $\hat{\mathbf{Y}}$ represents the downsampled prediction sequence.

# 4. Experiments

We conduct experiments to verify SpikF in time-series prediction tasks across multiple dimensions. Our evaluation framework mainly includes: (1) predictive performance on eight well-established long-sequence prediction benchmarks (2) computational efficiency through energy consumption analysis (3) individual component contributions via ablation studies of SPE, SFS and temporal synchronization mechanisms, and (4) generalization capability of the model for short-term prediction tasks and extended input sequences. The experimental results demonstrate that SpikF achieves superior accuracy and energy efficiency compared to SOTA models.

## 4.1. Long-term Prediction

In this section, we present the predictive performance of SpikF on well-established long-sequence prediction datasets.

**Datasets**  The experiments are conducted on eight widely-used long-term prediction datasets including ECL, Weather, ETT (ETTh1, ETTh2, ETTm1 and ETTm2), Traffic and Exchange, with detailed characteristics presented in Appendix B.1.

**Benchmarks**  We compare SpikF with three categories of advanced baseline models: Transformer-based models including Autoformer (Wu et al., 2022), Crossformer (Zhang & Yan, 2023), PatchTST (Nie et al., 2023), and iTransformer (Liu et al., 2024); TCN-based models such as SCINet (Liu et al., 2022) and TimesNet (Wu et al., 2023); and Linear-based models including DLinear and RLinear (Zeng et al., 2023). Notably, TimesNet (Wu et al., 2023) incorporates FFT into its model architecture.

**Main Results**  As shown in Table 1, SpikF demonstrates superior overall performance compared to other ANN-based models, achieving a $1.9\%$ reduction in prediction error compared to iTransformer in terms of MAE. Notably, on ETTm1 dataset, it achieves a $6.1\%$ lower MAE than iTransformer. While iTransformer focuses on correlations among different variates and PatchTST considers longer history information, both of them struggle to utilize positional information. In contrast, SpikF integrates patch encoding and S-FFT into its architecture, enabling utilization of local and global features while leveraging the sequential properties of time-series

data, which contribute to its superior accuracy over these benchmarks.

## 4.2. Model Analysis

**Computational Efficiency Analysis**  To evaluate the computational efficiency of our approach, we first analyze the variation in Synaptic Operations (SOPs) for S-FFT and S-iFFT as the firing rate $\alpha$ of the input sequence increases. Based on the theoretical analysis provided in Appendix A.3, the SOPs of S-FFT and S-iFFT operations for an input length of $L = m \times 2^n$ are:

$$\mathrm{SOPs}(S\text{-}FFT)$$
$$= \sum_{k=1}^{n} L \left[ 8\left(1 - \beta^{m \times 2^{k-1}}\right)^2 + 6\beta^{m \times 2^{k-1}} \left(1 - \beta^{m \times 2^{k-1}}\right) \right]$$
$$+ L\left(2m\alpha - 2 + 2\beta^m\right) \tag{15}$$

$$\mathrm{SOPs}(S\text{-}iFFT)$$
$$= \sum_{k=1}^{n} L \left[ 8\left(1 - \beta^{m \times 2^{k-1}}\right)^2 + 6\beta^{m \times 2^{k-1}} \left(1 - \beta^{m \times 2^{k-1}}\right) \right]$$
$$+ L\left(2m\alpha - 2 + 2\beta^m\right) + 6L \times m\alpha \tag{16}$$

where $\beta = 1 - \alpha$.

Figure 2 shows the formulas, indicating that the SOPs of S-FFT and S-iFFT are significantly lower than the Floating-Point Operations (FLOPs) of traditional FFT or iFFT when processing sparse input sequences. Since the energy consumption per SOP is $5.11$ times lower than that per FLOP (Yao et al., 2023), S-FFT in our model achieves superior energy efficiency compared to FFT in ANN models.

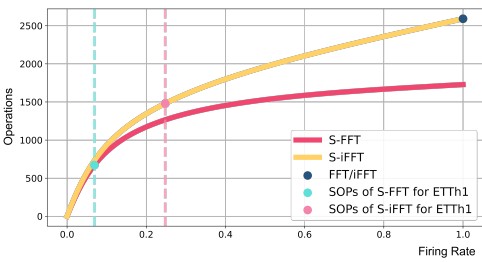

*Figure 2.* Theoretical computational complexity of the FFT and iFFT and their spiking version with respect to the variation of $\alpha$ for an input length of $L = 48$. SOPs of S-FFT and S-iFFT on ETTh1 dataset are illustrated in this figure.

To verify the efficiency of the SFS module, we compare it with common ANN-based temporal manipulation mechanisms in terms of the number of learnable parameters, operations (FLOPs or SOPs), and energy consumption caused

*Table 1.* In our multivariate forecasting analysis, we conduct extensive comparisons against multiple competitive models across varying prediction horizons, following the methodology established in iTransformer (Liu et al., 2024). All baseline models are configured with a uniform look-back window of $L = 96$ timesteps. Performance metrics are averaged over $H = \{96, 192, 336, 720\}$ prediction lengths. Lower MSE or MAE values indicate better performance. The best performance metrics are highlighted in **bold red**, while the second-best results are marked in underlined blue for clarity. This convention is consistently applied across all subsequent tables.

| Model | SpikF | | iTransformer | | RLinear | | PatchTST | | Crossformer | | TimesNet | | DLinear | | SCINet | | Autoformer | |
|---|---|---|---|---|---|---|---|---|---|---|---|---|---|---|---|---|---|---|
| Metric | MSE | MAE | MSE | MAE | MSE | MAE | MSE | MAE | MSE | MAE | MSE | MAE | MSE | MAE | MSE | MAE | MSE | MAE |
| ECL | 0.183 | 0.275 | **0.178** | **0.270** | 0.219 | 0.298 | 0.205 | 0.290 | 0.244 | 0.334 | 0.192 | 0.295 | 0.212 | 0.300 | 0.268 | 0.365 | 0.227 | 0.338 |
| Weather | **0.245** | **0.265** | 0.258 | 0.278 | 0.272 | 0.291 | 0.259 | 0.281 | 0.259 | 0.315 | 0.259 | 0.287 | 0.265 | 0.317 | 0.292 | 0.363 | 0.338 | 0.382 |
| ETTh1 | **0.440** | **0.428** | 0.454 | 0.447 | 0.446 | 0.434 | 0.469 | 0.454 | 0.529 | 0.522 | 0.458 | 0.450 | 0.456 | 0.452 | 0.747 | 0.647 | 0.496 | 0.487 |
| ETTh2 | **0.372** | **0.394** | 0.383 | 0.407 | 0.374 | 0.398 | 0.387 | 0.407 | 0.942 | 0.684 | 0.414 | 0.427 | 0.559 | 0.515 | 0.954 | 0.723 | 0.450 | 0.459 |
| ETTm1 | 0.388 | **0.385** | 0.407 | 0.410 | 0.414 | 0.407 | **0.387** | 0.400 | 0.513 | 0.496 | 0.400 | 0.406 | 0.403 | 0.407 | 0.485 | 0.481 | 0.588 | 0.517 |
| ETTm2 | **0.281** | **0.320** | 0.288 | 0.332 | 0.286 | 0.327 | **0.281** | 0.326 | 0.757 | 0.610 | 0.291 | 0.333 | 0.350 | 0.401 | 0.571 | 0.537 | 0.327 | 0.371 |
| Traffic | 0.497 | 0.296 | **0.428** | **0.282** | 0.626 | 0.378 | 0.481 | 0.304 | 0.550 | 0.304 | 0.620 | 0.336 | 0.625 | 0.383 | 0.804 | 0.509 | 0.628 | 0.379 |
| Exchange | 0.360 | **0.402** | 0.360 | 0.403 | 0.378 | 0.417 | 0.367 | 0.404 | 0.940 | 0.707 | 0.416 | 0.443 | **0.354** | 0.414 | 0.750 | 0.626 | 0.613 | 0.539 |

by operations per sample. Table 2 provides a comparative analysis of SFS with linear transform in the time domain (Zeng et al., 2023), linear transform in the frequency domain (Xu et al., 2024) and inverted self-attention (Liu et al., 2024). The results are summarized in Figure 2. The detailed calculation basis is provided in Appendix C.1.

As shown in Table 2, the SFS module, which leverages the grouped S-FFT paradigm and the event-driven nature of SNNs, reduces $42.66\%$ of the operational energy consumption compared to the time domain linear transform utilized by DLinear in terms of operational energy consumption.

*Table 2.* Comparison of the number of learnable parameters, operations, and operational energy consumption on the Electricity dataset with a look-back window of 96 and a prediction length of 720. The parameters listed in the table correspond to the key feature extraction module, which represents the Spiking Frequency Selector in SpikF, time-domain linear transform for DLinear, frequency-domain linear transform for FITS and inverted self-attention for iTransformer.

| Model | Paras | Opts | Energy/$\mu J$ |
|---|---|---|---|
| **SpikF** | **1.2K** | 0.13G | **117.66** |
| DLinear | 0.14M | 45M | 205.19 |
| FITS | 18K | 0.10G | 475.62 |
| iTransformer | 1.6M | 0.72G | 3289.29 |

To further underscore the energy efficiency of SpikF from a holistic perspective, we have compared the operational energy consumption of SpikF and iTransformer across all modules, including the encoder, feature extraction module, and decoder. As shown in Figure 3, SpikF with $T_s = 4$

reduces overall operational energy consumption by $75.05\%$ compared to iTransformer on the ECL dataset. Notably, operational energy consumption is reduced by $96.94\%$ for the encoder and $93.30\%$ for the feature extraction module.

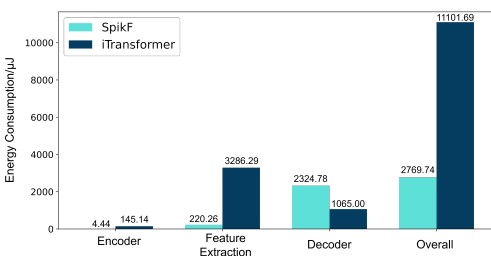

*Figure 3.* Systematic comparison of energy consumption between iTransformer and SpikF on ECL dataset, with a look-back window of 96 and a prediction length of 720. The comparison encompasses all modules, including the encoder, feature extraction, and decoder.

As indicated by (Lemaire et al., 2022; Shen et al., 2024), energy consumption comprises operational energy, memory-access energy, and addressing energy. To provide a more comprehensive analysis of SpikF's energy use, we follow the methods of (Lemaire et al., 2022; Shen et al., 2024) and present additional statistics on SpikF's energy efficiency in Table 3.

where ACE refers to S-ACE for iTransformer and NS-ACE for SpikF as proposed by (Shen et al., 2024), $E_{Mem}$ denotes the energy consumption for membrane accessing, $E_{Opts}$ refers to the energy consumption of operations, $E_{Addr}$ refers to the energy consumption for addressing, and $E_{Total}$ represents the total energy consumption.

*Table 3.* Energy consumption of SpikF and iTransformer.

| Model | ACE/M | $E_{Mem}/\mu J$ | $E_{Opts}/\mu J$ | $E_{Addr}/\mu J$ | $E_{Total}/\mu J$ |
|---|---|---|---|---|---|
| **SpikF** | $0.39 \downarrow_{6.27\times}$ | $1.14 \times 10^4$ | $2.77 \times 10^3$ | $4.34 \times 10^1$ | $1.37 \times 10^4 \downarrow_{3.16\times}$ |
| iTransformer | $2.47$ | $3.22 \times 10^4$ | $1.11 \times 10^4$ | $3.22 \times 10^{-1}$ | $4.34 \times 10^4$ |

In terms of ACE and $E_{Total}$, the energy consumption of SpikF is $6.27\times$ and $3.16\times$ lower than iTransformer respectively.

**Ablation Study** To evaluate the contributions of the three modules proposed in Section 3, we conduct an ablation study. For the SPE, we explore alternative encoding mechanisms, including convolution encoding and delta encoding. For the SFS, we replace the SFS module with inverted spiking self-attention (iSSA) as proposed in (Lv et al., 2024b), and we also examine the impact of completely discarding the SFS module. Additionally, for the decoder, we investigate the effect of removing the temporal synchronization mechanism. The implementation settings are provided in Appendix C.2. As demonstrated in Table 4, the SPE achieves $6.6\%$ and $3.2\%$ accuracy improvement over the Delta Encoder and Convolutional Encoder respectively, due to their limited utilization of long-term history information, while the SFS shows a $1.6\%$ performance enhancement compared to iSSA, for limitations of self-attention mechanisms as discussed in Appendix A.1. Additionally, temporal synchronization contributes a $0.5\%$ performance boost.

**Parameter Sensitivity Analysis** We evaluate the sensitivity of SpikF to temporal resolution $T_s$ by varying this parameter across five values 1, 2, 4, 8, and 16 while maintaining a fixed look-back window of 96 and a prediction horizon of 720 across all datasets. As illustrated in Figure 4, SpikF maintains robust performance across temporal resolution variations, with an average variation of only $3.7\%$. This highlights its ability to deliver consistent results even under reduced temporal resolution. Furthermore, these experimental results offer an approach for balancing accuracy and inference time, as lower temporal resolution typically corresponds to lower latency.

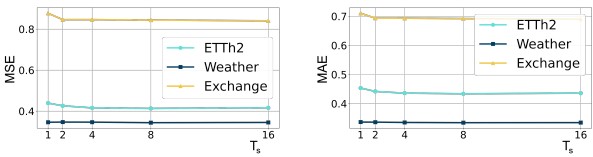

*Figure 4.* Performance of SpikF under varying temporal resolutions. Results show minimal fluctuations, indicating robustness to $T_s$ selection.

### 4.3. Model Generalizability

**Short-term Prediction** To demonstrate the versatility of SpikF, we extend our evaluation to the Solar-energy dataset, with detailed characteristics presented in Appendix B.1, specifically focusing on short-sequence prediction. This assessment aims to validate the model's applicability across diverse temporal scales and forecasting paradigms.

We select iTransformer (Liu et al., 2024) and iSpikformer (Lv et al., 2024b) as benchmarks, as they are SOTA ANN-based and SNN-based models, respectively, as indicated by (Lv et al., 2024b).

As demonstrated in Table 5, SpikF achieves lower prediction errors than both iSpikformer and iTransformer, with reductions of $2.3\%$ and $2.8\%$ respectively, showcasing the strengthened predictive capabilities of SpikF in handling both long-term and short-term prediction tasks.

*Table 5.* Short-term prediction results on the Solar-energy dataset. Following the experimental setup of (Lv et al., 2024b), the look-back window is fixed at 128, and the prediction lengths are set as $H = \{6, 24, 48, 96\}$. The Root Relative Squared Error (RSE) is used as the evaluation metric to compare the predictive performance of the models, where a lower RSE value indicates better performance. The performance results for iSpikformer and iTransformer are sourced from (Lv et al., 2024b).

| Model | Solar-energy | | | |
|---|---|---|---|---|
| | 6 | 24 | 48 | 96 |
| **SpikF** | **0.189** | 0.346 | **0.444** | **0.514** |
| iSpikformer | 0.204 | **0.333** | 0.465 | 0.521 |
| iTransformer | 0.191 | 0.348 | 0.448 | 0.563 |

**Extension of Input Sequence** It is widely believed that model performance should improve with longer input sequences, as incorporating more historical information enhances the model's ability to capture temporal patterns. Therefore, the ability to utilize longer historical information serves as an indicator of a model's generalizability and predictive performance. To explore this, we modify the look-back window across five temporal spans: 48, 96, 192, 336, and 720 time steps, while simultaneously adjusting the prediction length from 96, 192, 336, to 720 time steps. As shown in Figure 5, the loss values generally decrease as the input sequence length increases across all prediction lengths. Specifically, when the length of input sequence ex-

*Table 4.* Ablation study results of the aforementioned strategies. The full ablation study results are available in Appendix C.2.

| Module | Design | ETTh1 | | ETTm1 | | Weather | |
|---|---|---|---|---|---|---|---|
| | | MSE | MAE | MSE | MAE | MSE | MAE |
| **SpikF** | original | **0.440** | **0.428** | **0.388** | **0.385** | **0.245** | **0.265** |
| Spike Encoder | Delta | 0.468 | 0.450 | 0.431 | 0.424 | 0.260 | 0.281 |
| | Conv | 0.467 | 0.446 | 0.404 | 0.397 | 0.249 | 0.272 |
| Feature Extraction | w/o | 0.441 | **0.428** | **0.388** | 0.387 | 0.246 | 0.266 |
| | iSSA | 0.452 | 0.440 | 0.389 | 0.389 | 0.247 | 0.268 |
| Spike Decoder | w/o | 0.444 | 0.430 | 0.391 | 0.387 | 0.246 | 0.266 |

pands from 48 to 720, an averaged prediction promotion is 9.0% in terms of MSE and 2.4% in terms of MAE, demonstrating the model's effective utilization of longer history information.

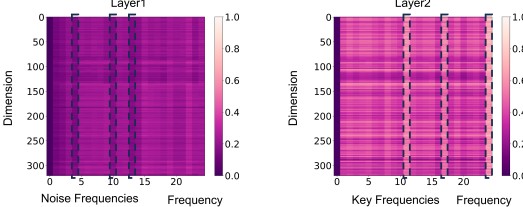

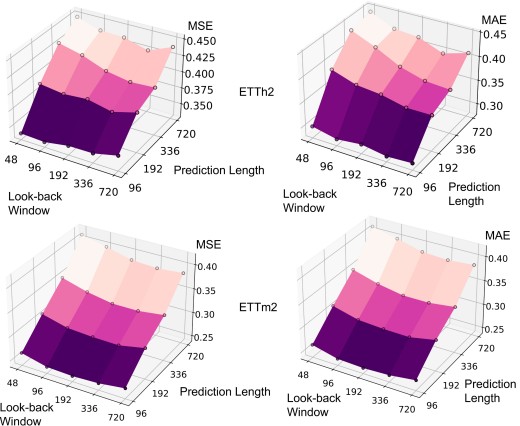

*Figure 5.* MSE and MAE values for the ETTh2 (top) and ETTm2 (bottom) datasets across varying input and output lengths. The results highlight our model's generalization capability, as evidenced by the consistent decrease in MSE and MAE values with the increase of look-back window.

*Figure 6.* Visualization of the SFS module across different layers. The data is obtained from the ECL dataset with a look-back window of 96 timesteps and a prediction horizon of 720 timesteps. The left panel shows the selection probabilities for the first SFS module, while the right panel corresponds to the second module. The noise frequency components and key frequency components is highlighted in this figure.

### 4.4. Visualization

To validate the feature extraction capability of the SFS module, we plot the averaged selection matrix $\mathbf{M}_{sel}$ for different SFS layers on the ECL dataset, as shown in Figure 6.

In Figure 6, deeper SFS module tends to discard more frequency components, while shallower module focuses on selecting useful frequencies. This is because that the first SFS module enhances the predictive utilization of input spike trains by selecting relevant frequency modes. As a result, the input to the second module contains less noise, leading to a higher tendency for the second SFS module to retain more frequency components. This analysis highlights the necessity and effectiveness of the design of SFS.

## 5. Conclusion

In this study, we introduce SpikF, a novel architecture designed for long-term prediction tasks, bridging the critical gap of SNN benchmarks in this domain. The Spiking Patch Encoder enables efficient encoding of extended historical information, while the Spiking Frequency Selector module inherently resolves the challenge of utilizing positional information in SNN architectures and facilitates global feature extraction. This work not only significantly broadens the potential applications of SNN architectures but also offers an energy-efficient solution for the time-series prediction community.

## Acknowledgements

This work is supported by National Natural Science Foundation of China (No. 62334014), and the authors gratefully acknowledge the insightful comments and constructive suggestions provided by the editor and the anonymous reviewers, which have substantially enhanced the clarity, rigor, and overall quality of this manuscript.

## Impact Statement

This paper presents work whose goal is to advance the field of Machine Learning. There are many potential societal consequences of our work, none which we feel must be specifically highlighted here.

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

# A. Theoretical Analysis

## A.1. Permutation-Invariance of Self-attention and Inverted Self-attention Mechanisms

Consider an input sequence $\mathbf{X} \in \mathbb{R}^{L \times D}$, where $\mathbf{X}_{i \leftrightarrow j,.}$ denotes $\mathbf{X}$ with its $i$th and $j$th rows exchanged, and $\mathbf{X}_{.,i \leftrightarrow j}$ denotes $\mathbf{X}$ with its $i$th and $j$th columns exchanged. For the standard self-attention mechanism, the queries $\mathbf{Q}$, keys $\mathbf{K}$, and values $\mathbf{V}$ are computed as follows:

$$\mathbf{Q} = \mathbf{X}\mathbf{W}_Q, \quad \mathbf{K} = \mathbf{X}\mathbf{W}_K, \quad \mathbf{V} = \mathbf{X}\mathbf{W}_V \tag{17}$$

When the rows of $\mathbf{X}$ are permuted, the corresponding transformations yield:

$$\mathbf{X}_{i \leftrightarrow j,.}\mathbf{W}_Q = \mathbf{Q}_{i \leftrightarrow j,.}, \quad \mathbf{X}_{i \leftrightarrow j,.}\mathbf{W}_K = \mathbf{K}_{i \leftrightarrow j,.}, \quad \mathbf{X}_{i \leftrightarrow j,.}\mathbf{W}_V = \mathbf{V}_{i \leftrightarrow j,.} \tag{18}$$

Furthermore, the attention scores and output are permuted accordingly:

$$\mathbf{Q}_{i \leftrightarrow j,.}\mathbf{K}_{i \leftrightarrow j,.}^T = (\mathbf{Q}\mathbf{K}^T)_{i \leftrightarrow j, i \leftrightarrow j} \tag{19}$$

and

$$\text{SoftMax}\left(\frac{\mathbf{Q}_{i \leftrightarrow j,.}\mathbf{K}_{i \leftrightarrow j,.}^T}{\sqrt{D}}\right)\mathbf{V}_{i \leftrightarrow j,.} = \left(\text{SoftMax}\left(\frac{\mathbf{Q}\mathbf{K}^T}{\sqrt{D}}\right)\mathbf{V}\right)_{i \leftrightarrow j,.} \tag{20}$$

For the inverted self-attention mechanism, where $\mathbf{X} \in \mathbb{R}^{D \times L}$, the formulas are:

$$\mathbf{X}_{.,i \leftrightarrow j}(\mathbf{W}_Q)_{i \leftrightarrow j,.} = \mathbf{Q}, \quad \mathbf{X}_{.,i \leftrightarrow j}(\mathbf{W}_K)_{i \leftrightarrow j,.} = \mathbf{K}, \quad \mathbf{X}_{.,i \leftrightarrow j}(\mathbf{W}_V)_{i \leftrightarrow j,.} = \mathbf{V} \tag{21}$$

This indicates that permuting the input $\mathbf{X}$ along the temporal dimension leads to a corresponding permutation of the output attention map. As a result, both the standard self-attention mechanism and its inverted variant are unable to effectively capture positional information. In the absence of subsequent position-sensitive operations, the model's prediction output would remain invariant under the assumption that the optimization process is guaranteed to avoid local minima. Similarly, the spiking versions of self-attention and inverted self-attention exhibit insensitivity to the relative position of the input sequence due to the same limitation.

In contrast, our proposed Spiking Frequency Selector (SFS) module is inherently sensitive to the relative positions of input spikes, owing to the inclusion of the S-FFT operation. This property enables the SFS module to better model temporal dependencies.

To experimentally evaluate the utilization of positional information, we uniformly shuffled the input sequence and compared the prediction accuracy with that of the ordered input sequence. As shown in Figure 7, SpikF achieves a $0.8\%$ improvement in MAE when sequential properties are incorporated, while iTransformer shows no performance improvement, highlighting its ineffective utilization of sequential properties.

## A.2. The Role of Fourier Transform in SNN Architectures

As (Lv et al., 2024b) suggests, the dynamics of membrane potential in SNNs provide a unique method for capturing temporal data intricacies. However, this can result in a separated receptive field, potentially missing global temporal information.

**Proof**:

Given the dynamics of LIF neurons:

$$U[t] = V[t-1] + \frac{1}{\tau_m}\left(I[t] - V[t-1] + V_{rest}\right) \tag{22}$$

$$S[t] = H\left(U[t] - V_{th}\right) \tag{23}$$

$$V[t] = U[t]\left(1 - S[t]\right) + V_{rest}S[t] \tag{24}$$

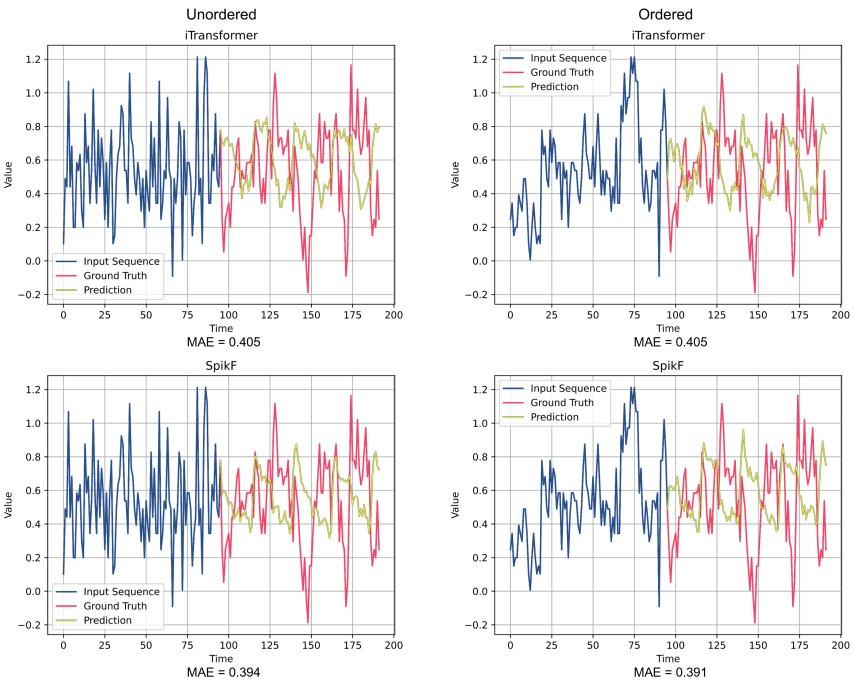

*Figure 7.* Comparison of model performance with unordered versus ordered input sequences.

For two series of stimulation $I_1[1], I_1[2], ..., I_1[t^*]$ and $I_2[1], I_2[2], ..., I_2[t^*]$ where $S[1] = S[2] = ... = S[t^* - 1] = 0$ and $S[t^*] = 1$, these sequences are equivalent in terms of membrane potential when $t \geq t^*$, as $U[t^*] = V_{rest}$.

If we assume that $S[t^1] = S[t^2] = ... = S[t^s] = 1$ and $S[t] = 0$ otherwise. Then the receptive field of the LIF neuron is limited to the regions $[1, t^1], [t^1 + 1, t^2], ..., [t^{s-1} + 1, t^s]$ and $[t^s + 1, T]$.

This limitation hinders high-prediction SNNs in long-term prediction domains, which require modeling long-term dependencies (Zeng et al., 2023). Thus, relying solely on SNN internal dynamics is insufficient; external dynamics are necessary for modeling long-term dependencies.

While typical methods to expand the receptive field of SNN involve linear layers and self-attention mechanisms, these are less suitable for sequential tasks due to their permutation-invariance (Zeng et al., 2023), which has been proved in Appendix A.1.

In contrast, FFT transforms temporal series into the frequency domain, expanding the receptive field to the entire time-series. Modifications in the frequency domain influence the entire series, and sequential information is inherently embedded in frequency components via FFT's rotation factors:

$$F[k] = \sum_{t=1}^{T} S[t] e^{-j \frac{2\pi}{T} kt} \tag{25}$$

Thus, selecting frequency components allows for global influence, as $F[k]$ is a function of $S[1], S[2], ..., S[T]$, thus making FFT an ideal approach for external dynamics.

In summary, incorporating FFT into SNN architecture is essential in the time-series domain to expand the receptive field and improve long-term dependency modeling.

### A.3. Computational Complexity of S-FFT and S-iFFT Algorithm

While numerous optimizations have been proposed to enhance the computational efficiency of Fast Fourier Transform (FFT), we give a theoretical analysis based on the classical FFT algorithm and its spiking version, as shown in Algorithm 1, due to its fundamental nature and universal applicability.

---
**Algorithm 1** Fast Fourier Transform

---
**Require:** Input sequence $x[t]$ of length $L = m \times 2^n$, where $m$ is not divisible by 2
**Ensure:** FFT output $X[k]$
 1: {Define a recursive procedure for FFT}
 2: **Procedure** RecursiveFFT$(x, m, n)$
 3: **if** $n = 0$ **then**
 4:    {Base case: Directly compute DFT for segments of length $m$}
 5:    $X \leftarrow \text{DFT}(x, m)$
 6:    **return** $X$
 7: **else**
 8:    {Divide the sequence into even and odd parts}
 9:    $L \leftarrow m \times 2^n$
10:    $x_{\text{even}} \leftarrow [x[0], x[2], \ldots, x[L-2]]$
11:    $x_{\text{odd}} \leftarrow [x[1], x[3], \ldots, x[L-1]]$
12:    {Recursively compute FFT for even and odd parts}
13:    $X_{\text{even}} \leftarrow \text{RecursiveFFT}(x_{\text{even}}, m, n-1)$
14:    $X_{\text{odd}} \leftarrow \text{RecursiveFFT}(x_{\text{odd}}, m, n-1)$
15:    {Combine the results using the Cooley-Tukey butterfly}
16:    **for** $k \leftarrow 0$ to $L/2 - 1$ **do**
17:       $t \leftarrow W_L^k \times X_{\text{odd}}[k]$
18:       $X[k] \leftarrow X_{\text{even}}[k] + t$
19:       $X[k + L/2] \leftarrow X_{\text{even}}[k] - t$
20:    **end for**
21:    **return** $X$
22: **end if**
23: {Main call to the recursive procedure}
24: $X \leftarrow \text{RecursiveFFT}(x, m, n)$
25: **return** $X$

---

where $W_{\text{L}}^k = e^{\frac{-2\pi i k}{\text{L}}}$ denotes the twiddle factor.

For S-FFT, we assume that each position of the input sequence fires randomly with a rate of $\alpha$. This hypothesis remains valid despite the temporal dynamics of the model, as it is supported by the reordering in the butterfly algorithm. In the following analysis, a complex addition is considered as 2 SOPs (addition of real and imaginary parts), and a complex multiplication is considered as 6 SOPs (4 multiplications and 2 additions). The calculation formulas of SOPs are provided in Appendix C.1.

In the first stage, when the algorithm reaches the recursive base case of S-FFT, the spiking components are aggregated to perform a traditional S-DFT transformation, which calculates the transformed sequence within the group. The number of SOPs in this stage is given by:

$$\sum_{k=1}^{m} 2 \times 2^n \times m\,(k-1)\,P_k = L\,(2m\alpha - 2 + 2\beta^m) \tag{26}$$

where $P_k$ represents the probability of exactly $k$ neurons within the group spiking, and $\beta = 1 - \alpha$ denotes the non-spiking ratio.

In the second stage, when the odd and even sequences are used to compute the final S-FFT results, the following rules apply for each component:

- If the corresponding odd S-FFT component equals zero, no calculation is performed.

- If the odd component is non-zero and the corresponding even component is zero, only one complex multiplication is performed.

- Otherwise, one complex multiplication and one complex addition are performed.

Thus, the number of SOPs in the second stage is:

$$
\begin{aligned}
&\sum_{k=1}^{n} 2^{n-k} \times m \times 2^k \left[ 8\left(1 - \beta^{m\times 2^{k-1}}\right)^2 + 6\beta^{m\times 2^{k-1}}\left(1 - \beta^{m\times 2^{k-1}}\right)\right] \\
&= \sum_{k=1}^{n} L \left[ 8\left(1 - \beta^{m\times 2^{k-1}}\right)^2 + 6\beta^{m\times 2^{k-1}}\left(1 - \beta^{m\times 2^{k-1}}\right)\right]
\end{aligned}
\tag{27}
$$

Therefore, the total SOPs for S-FFT are:

$$
\mathrm{SOPs}(S\text{-}FFT) = \sum_{k=1}^{n} L \left[ 8\left(1 - \beta^{m\times 2^{k-1}}\right)^2 + 6\beta^{m\times 2^{k-1}}\left(1 - \beta^{m\times 2^{k-1}}\right)\right] + L\left(2m\alpha - 2 + 2\beta^m\right)
\tag{28}
$$

For S-iFFT, the input sequence is real-valued, though many elements are zero due to the selection mechanism. Thus, in the first stage, additional calculations involving the input sequence multiplied by the rotation factor are required. By adding this part, the SOPs for S-iFFT are:

$$
\mathrm{SOPs}(S\text{-}iFFT) = \sum_{k=1}^{n} L \left[ 8\left(1 - \beta^{m\times 2^{k-1}}\right)^2 + 6\beta^{m\times 2^{k-1}}\left(1 - \beta^{m\times 2^{k-1}}\right)\right] + L\left(2m\alpha - 2 + 2\beta^m\right) + 6L \times m\alpha
\tag{29}
$$

It is important to note that S-FFT and S-iFFT essentially transform real-number multiplication into inter-synaptic spike transitions and real-number addition into the accumulation of stimulation from different pre-synaptic neurons. Therefore, the above analysis on the SOPs of S-iFFT can be converted into FLOPs of FFT and iFFT by setting $\alpha = 1$. Thus, the FLOPs for FFT and iFFT operations are:

$$
\mathrm{FLOPs}(FFT) = \mathrm{FLOPs}(iFFT) = L\left(8m + 8n - 2\right)
\tag{30}
$$

### A.4. The Role of Average Pooling

Under the assumption that the prediction errors independently follow the Laplace distribution, which is the theoretical foundation of the MAE loss function, we establish the following mathematical framework to analyze the contribution of the average pooling layer in the MLP Decoder. Let the prediction errors before average pooling be denoted as $e_1, e_2, \ldots, e_{T_s}$, where each $e_k$ is independently and identically distributed (i.i.d.) according to the Laplace distribution:

$$
e_1, e_2, \ldots, e_{T_s} \text{ i.i.d. } e_1 \sim f(x) = \frac{1}{2}e^{-|x|}
\tag{31}
$$

The characteristic function of the Laplace distribution is given by:

$$
\phi(t) = \frac{1}{1 + t^2}
\tag{32}
$$

Consider the error after applying average pooling, defined as $e = \frac{1}{T_s}\sum_{k=1}^{T_s} e_k$. The characteristic function of $e$ can be expressed as:

$$
\phi_e(t|T_s) = \frac{1}{\left(1 + \frac{t^2}{T_s^2}\right)^{T_s}}
\tag{33}
$$

Using the characteristic function inversion method, the probability density function of $e$ is derived as:

$$
f_e(x|T_s) = \frac{1}{2\pi}\int_{-\infty}^{\infty} \frac{\exp(-itx)}{\left(1 + \frac{t^2}{T_s^2}\right)^{T_s}} \, dt
\tag{34}
$$

The expectation of the absolute error $\mathbb{E}_e[|x|](T_s)$ is given by:

$$\mathbb{E}_e[|x|](T_s) = \int_{-\infty}^{\infty} \frac{|x|}{2\pi} \int_{-\infty}^{\infty} \frac{\exp(-itx)}{\left(1 + \frac{t^2}{T_s^2}\right)^{T_s}} \, dt \, dx = \frac{2}{\sqrt{\pi}} \cdot \frac{\Gamma\left(T_s + \frac{1}{2}\right)}{\Gamma(T_s + 1)} \tag{35}$$

where $\Gamma(z) = \int_0^{\infty} t^{z-1} e^{-t} \, dt$.

When $T_s$ is incremented to $T_s + 1$, the expectation of the absolute error decreases by a factor of $\frac{T_s + \frac{1}{2}}{T_s + 1}$. This relationship is expressed as:

$$\frac{\mathbb{E}_e[|x|](T_s + 1)}{\mathbb{E}_e[|x|](T_s)} = \frac{T_s + \frac{1}{2}}{T_s + 1} \tag{36}$$

It is important to note that as $T_s$ approaches infinity, the expectation of the absolute error converges to zero:

$$\lim_{T_s \to +\infty} \mathbb{E}_e[|x|](T_s) = 0 \tag{37}$$

Above analysis implies that when $T_s$ increases, the expectation of MAE will decrease because of the average pooling layer.

## B. Experimental Details

### B.1. Depiction of the Datasets

In our experimental evaluation, we utilize several well-established real-world datasets that have been extensively validated in prior time-series forecasting research. The **Electricity** dataset (Wu et al., 2022) records hourly interval electricity consumption data from 321 clients. The **Weather** dataset (Wu et al., 2022) includes meteorological measurements collected every 10 minutes from the Weather Station of the Max Planck Biogeochemistry Institute in 2020, containing 21 variables. The **ETT** dataset (Zhou et al., 2021) consists of four subsets: ETTh1 and ETTh2 with hourly interval data, and ETTm1 and ETTm2 with 15-minute interval data, each containing two years (2016-2018) of measurements from electrical transformers, including variables like oil temperature and load. The **Traffic** dataset (Wu et al., 2022) provides hourly road occupancy rates from 862 sensors in the San Francisco Bay area between 2015 and 2016. The **Exchange** dataset (Wu et al., 2022) comprises daily exchange rate data from eight countries spanning from 1990 to 2016. **Solar-Energy** (Lai et al., 2018), provides a dataset documenting the solar power generation of 137 photovoltaic plants throughout the year 2006, with measurements recorded at 10-minute intervals. All datasets are preprocessed to handle missing values and normalized to ensure consistent scaling across variables, following the approach of (Liu et al., 2024). These datasets represent diverse domains with varying temporal patterns, seasonality, and trends, making them ideal benchmarks for evaluating the robustness and generalizability of time-series forecasting models.

We fix the look-back window for all datasets except Solar-energy to 96, consistent with iTransformer (Liu et al., 2024) for long-term prediction tasks. The look-back window of Solar-energy is set to 128, following (Lv et al., 2024b). The train-validation-test split method follow the iTransformer (Liu et al., 2024) and iSpikformer (Lv et al., 2024b) approach to ensure fairness. Detailed dataset information is provided in Table 6.

### B.2. Metrics

For long-term forecasting tasks, the Mean Absolute Error (MAE) and Mean Squared Error (MSE) are employed as evaluation metrics, defined as follows:

$$\text{MAE} = \|\mathbf{Y} - \hat{\mathbf{Y}}\| \tag{38}$$

$$\text{MSE} = \|\mathbf{Y} - \hat{\mathbf{Y}}\|^2 \tag{39}$$

where $\mathbf{Y}$ and $\hat{\mathbf{Y}}$ represent the ground truth and predicted values, respectively.

For short-term forecasting tasks, the Root Relative Squared Error (RSE) is utilized to assess model performance by (Lv et al., 2024b). The metric is defined as:

*Table 6.* Dataset descriptions. The dataset size is organized in (Train, Validation, Test).

| Task | Dataset | Dim | Sampling Frequency | Dataset Size | Depiction |
|------|---------|-----|--------------------|--------------|-----------|
| Long | Electricity | 321 | 1 Hour | (18317, 2633, 5261) | Electricity |
| | Weather | 21 | 10 Minutes | (36792, 5271, 10540) | Weather |
| | ETTh | 7 | 1 Hour | (8545, 2881, 2881) | Electricity |
| | ETTm | 7 | 15 Minutes | (34465, 11521, 11521) | Electricity |
| | Traffic | 862 | 1 Hour | (12185, 1757, 3509) | Transportation |
| | Exchange | 8 | 1 Day | (5120, 665, 1422) | Economy |
| Short | Solar-energy | 137 | 10 Minutes | (31362, 10506, 10506) | Energy |

$$\text{RSE} = \sqrt{\frac{\sum_{i=1}^{B} \|\mathbf{Y}_i - \hat{\mathbf{Y}}_i\|^2}{\sum_{i=1}^{B} \|\mathbf{Y}_i - \bar{\mathbf{Y}}\|^2}} = \sqrt{\frac{\sum_{i=1}^{B} \sum_{j=1}^{H} \sum_{k=1}^{D} (Y_{i,j,k} - \hat{Y}_{i,j,k})^2}{\sum_{i=1}^{B} \sum_{j=1}^{H} \sum_{k=1}^{D} (Y_{i,j,k} - \bar{Y}_{.,j,k})^2}} \tag{40}$$

where $B$ denotes the number of samples, $H$ represents the prediction length, and $D$ corresponds to the number of dimensions.

### B.3. Implementation Details

All experiments are built on the frameworks developed on PyTorch (Paszke et al., 2019) and SpikingJelly (Fang et al., 2023), the latter being an SNN repository built upon PyTorch. SpikingJelly has been rigorously validated in prior studies, including Spikformer (Zhou et al., 2023) and QKFormer (Zhou et al., 2024). The experiments are executed on a single NVIDIA 3090 GPU.

The surrogate function for spatio-temporal backpropagation (STBP) is selected as the Sigmoid function, defined as:

$$\sigma(x) = \frac{1}{1 + e^{-\alpha x}}, \tag{41}$$

where the parameter $\alpha$ is set to $4.0$, consistent with previous studies (Zhou et al., 2023; 2024) across all experiments.

Figure 8 provides a visual comparison between the Heaviside step function and the Sigmoid function, along with the derivative of the Sigmoid function.

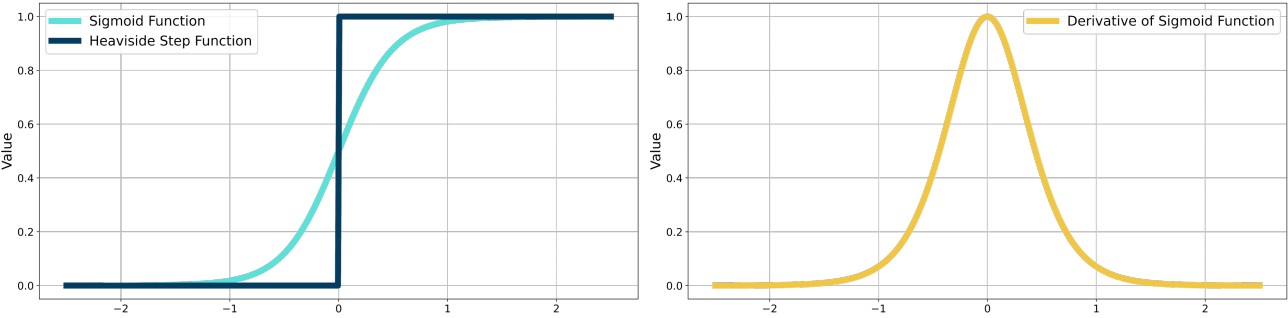

*Figure 8.* Comparison of the Heaviside step function and the Sigmoid function (left), and visualization of the derivative of the Sigmoid function (right).

The parameters for the spiking neurons are predominantly set to their default values within the SpikingJelly library, with the time constant $\tau_m$ fixed at $2.0$ and the threshold potential $v_{th}$ at $1.0$. An exception is made for the generator LIF neurons, where $v_{th}$ was adjusted to $0.1$ to improve passing rate of the selector. The Adam optimizer is employed with its parameters configured as follows: $\beta_1 = 0.9$ and $\beta_2 = 0.999$.

The temporal resolution $T_s$ is set to 16 for datasets exhibiting minimal variables and 4 for those with greater variables. The feature extraction layers of the architecture are chosen from the set $\{1, 2, 4\}$, and the patch dimension are selected from

$\{8, 16, 32, 64\}$. The hidden layer dimension within the encoder's MLP are chosen from $\{180, 360, 540, 720\}$. Batch sizes are selected from $\{4, 8, 16, 32, 64\}$, and initial learning rates are set from $\{10^{-4}, 5 \times 10^{-4}, 10^{-3}\}$. The number of training epochs is chosen from $\{5, 10, 15, 20\}$. An early-stopping strategy is implemented to optimize the use of the validation set for model evaluation. All model parameters are fine-tuned based on validation set performance. A comprehensive tabulation of parameter selections is provided in Table 7.

*Table 7.* Detailed hyperparameter selection of long term prediction datasets.

| Dataset | $T_s$ | Layers | $d_{patch}$ | $d_{hidden}$ | Batch Size | lr | Epoches |
|---|---|---|---|---|---|---|---|
| Electricity | 4 | 2 | 16 | 720 | 8 | $5 \times 10^{-4}$ | 15 |
| Weather | 16 | 2 | 32 | 360 | 32 | $10^{-3}$ | 5 |
| ETTh | 16 | 1 | 32 | 720 | 32 | $5 \times 10^{-4}$ | 5 |
| ETTm | 16 | 1 | 32 | 720 | 32 | $5 \times 10^{-4}$ | 10 |
| Traffic | 4 | 1 | 16 | 540 | 4 | $5 \times 10^{-4}$ | 15 |
| Exchange | 16 | 1 | 32 | 720 | 32 | $10^{-4}$ | 10 |
| Solar-Energy | 4 | 2 | 16 | 360 | 16 | $10^{-3}$ | 5 |

## C. Model Analysis

### C.1. Model Efficiency

For SNN models, the SOPs of each layer are given by:

$$\text{SOPs} = \alpha \times T \times \text{FLOPs} \tag{42}$$

where $\alpha$ represents the firing rate of input spike trains, $T$ is the simulation time steps of the SNN, and FLOPs denotes the floating-point operations of the layer.

For the exact power cost of our SFS module, as detailed in Table 2, we assume the operations are implemented on 45nm hardware (Yao et al., 2023; Zhou et al., 2023), with an energy consumption of 4.6 pJ per FLOP and 0.9 pJ per SOP.

### C.2. Ablation Study

To evaluate the contribution of each module in SpikF's design to its overall performance, we conduct detailed ablation studies. These experiments involve systematically replacing or removing individual components while keeping the rest of the model unchanged.

For spike encoder, we choose Delta Encoder which can be formulated as:

$$\mathbf{H} = \text{Diff}(\mathbf{X}) \tag{43}$$

$$\mathbf{S}_{enc} = \mathcal{SN}(\text{ZOH}(\text{BN}(\mathbf{H}))) \tag{44}$$

where Diff represents the differential operation, and Convolutional Encoder:

$$\mathbf{H} = \text{Conv}(\mathbf{X}) \tag{45}$$

$$\mathbf{S}_{enc} = \mathcal{SN}(\text{ZOH}(\text{BN}(\mathbf{H}))) \tag{46}$$

where Conv denotes the convolutional layer.

For feature extraction module, we consider removing the SFS and replacing it with inverted spiking self-attention mechanism which is proposed by (Lv et al., 2024b) which can be expressed as:

$$\mathbf{Q} = \mathcal{SN}_Q(\text{BN}(\mathbf{S}_{enc}\mathbf{W}_Q)), \quad \mathbf{K} = \mathcal{SN}_K(\text{BN}(\mathbf{S}_{enc}\mathbf{W}_K)), \quad \mathbf{V} = \mathcal{SN}_V(\text{BN}(\mathbf{S}_{enc}\mathbf{W}_V)) \tag{47}$$

$$\mathbf{Attn} = \mathbf{QK}^T\mathbf{V} \times scaler \tag{48}$$

where $scaler$ is a pre-defined constant. For spike decoder module, we consider removing the temporal synchronization mechanism.

Full results of our ablation study are shown at Table 8.

*Table 8.* Comprehensive results of the ablation study.

| Module | Design | Prediction Length | ETTh1 MSE | ETTh1 MAE | ETTm1 MSE | ETTm1 MAE | Weather MSE | Weather MAE |
|---|---|---|---|---|---|---|---|---|
| SpikF | original | 96 | **0.379** | 0.391 | **0.317** | **0.345** | **0.163** | **0.200** |
| | | 192 | **0.432** | **0.421** | 0.372 | **0.372** | **0.209** | **0.241** |
| | | 336 | 0.473 | **0.441** | 0.401 | **0.394** | 0.266 | **0.283** |
| | | 720 | **0.474** | **0.459** | **0.461** | **0.430** | **0.344** | **0.334** |
| | | Avg | **0.440** | **0.428** | **0.388** | **0.385** | **0.245** | **0.265** |
| Spike Encoder | Delta | 96 | 0.419 | 0.422 | 0.372 | 0.393 | 0.179 | 0.219 |
| | | 192 | 0.463 | 0.445 | 0.421 | 0.417 | 0.225 | 0.259 |
| | | 336 | 0.502 | 0.462 | 0.438 | 0.429 | 0.279 | 0.297 |
| | | 720 | 0.487 | 0.470 | 0.493 | 0.456 | 0.357 | 0.347 |
| | | Avg | 0.468 | 0.450 | 0.431 | 0.424 | 0.260 | 0.289 |
| | Conv | 96 | 0.406 | 0.411 | 0.343 | 0.369 | 0.166 | 0.210 |
| | | 192 | 0.457 | 0.437 | 0.382 | 0.383 | 0.214 | 0.250 |
| | | 336 | 0.502 | 0.459 | 0.409 | 0.399 | 0.270 | 0.290 |
| | | 720 | 0.502 | 0.475 | 0.480 | 0.436 | 0.346 | 0.338 |
| | | Avg | 0.467 | 0.446 | 0.404 | 0.397 | 0.249 | 0.272 |
| Feature Extraction | w/o | 96 | 0.380 | **0.390** | **0.317** | 0.350 | 0.164 | 0.201 |
| | | 192 | 0.434 | **0.421** | 0.369 | **0.372** | 0.210 | 0.243 |
| | | 336 | **0.472** | **0.441** | **0.399** | 0.395 | 0.266 | 0.284 |
| | | 720 | 0.476 | 0.462 | 0.468 | **0.430** | 0.345 | 0.336 |
| | | Avg | 0.441 | **0.428** | **0.388** | 0.387 | 0.246 | 0.266 |
| | iSSA | 96 | 0.389 | 0.402 | 0.324 | 0.350 | 0.166 | 0.203 |
| | | 192 | 0.436 | 0.428 | **0.366** | 0.374 | 0.213 | 0.245 |
| | | 336 | 0.488 | 0.454 | 0.400 | 0.396 | 0.267 | 0.287 |
| | | 720 | 0.495 | 0.476 | 0.465 | 0.435 | **0.344** | 0.338 |
| | | Avg | 0.452 | 0.440 | 0.389 | 0.389 | 0.247 | 0.268 |
| Spike Decoder | w/o | 96 | 0.383 | 0.391 | 0.319 | 0.346 | **0.163** | **0.200** |
| | | 192 | 0.435 | 0.422 | 0.374 | 0.373 | 0.211 | 0.243 |
| | | 336 | 0.483 | 0.445 | 0.407 | 0.398 | **0.265** | **0.283** |
| | | 720 | 0.478 | 0.461 | 0.466 | 0.433 | 0.346 | 0.336 |
| | | Avg | 0.444 | 0.430 | 0.391 | 0.387 | 0.246 | 0.266 |

## C.3. Sensitivity to Patch Dimension

To evaluate SpikF's sensitivity to patch dimension selection, we conduct experiments with varying patch dimension configurations $\{4, 8, 16, 32\}$ across the ETTh1, ETTm1, and ETTm2 datasets. Using a fixed look-back window of 96 time steps and a prediction horizon of 720 time steps, our analysis shows minimal performance fluctuations across different patch dimension configurations, with an average variation of $1.2\%$. Notably, the ETTm2 dataset demonstrates exceptional stability, exhibiting only $0.5\%$ variation. This consistent behavior demonstrates SpikF's robust performance relative to patch dimension selection in the encoding process.

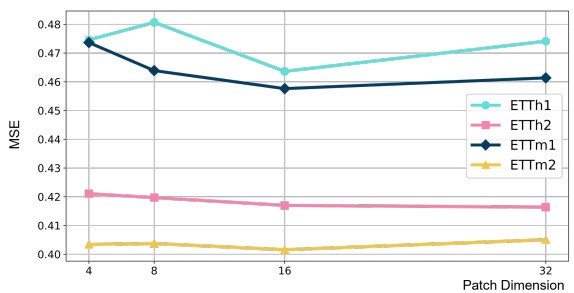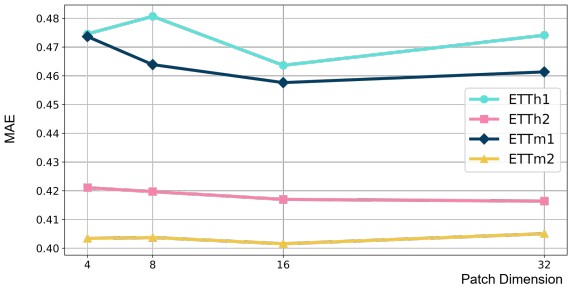

*Figure 9.* Sensitivity to patch dimension selection.

## C.4. Error Bars

To assess the robustness of our proposed model, we perform three independent experimental trials with distinct random seeds and calculate the corresponding 95% confidence intervals, as detailed in Table 9 and Table 10. It is worth noting that SpikF is optimized using the MAE loss function, in contrast to the widely adopted MSE loss function for other benchmark models. Therefore, to ensure a more fair comparison, we also include the 95% confidence intervals for the iTransformer model using the Mean Absolute Error (MAE) loss function. As evidenced in Table 9 and Table 10, SpikF not only outperforms iTransformer in prediction accuracy but also demonstrates significantly improved robustness. Specifically, on the ETTm2, Weather, and Exchange datasets, SpikF achieves an average confidence interval length which is 85.1% shorter than that of iTransformer.

*Table 9.* 95% confidence intervals of SpikF and iTransformer on ETTh1, ETTh2, and ETTm1 datasets with MAE loss function. Better performance is highlighted in **bold**.

| Model | | ETTh1 | | ETTh2 | | ETTm1 | |
|---|---|---|---|---|---|---|---|
| Metric | | MSE | MAE | MSE | MAE | MSE | MAE |
| SpikF | 96 | **0.379**±**0.003** | **0.391**±**0.001** | **0.290**± 0.001 | **0.336**±0.003 | **0.317**±0.005 | **0.345**±0.003 |
| | 192 | **0.432**±0.008 | **0.421**±**0.002** | **0.367**±0.006 | **0.385**±**0.001** | **0.372**±0.007 | **0.372**±0.005 |
| | 336 | **0.473**±**0.010** | **0.441**±**0.002** | **0.414**±0.011 | **0.420**±0.007 | **0.401**±0.002 | **0.394**±0.004 |
| | 720 | **0.474**±0.016 | **0.459**±**0.006** | **0.416**±0.008 | **0.436**±**0.001** | **0.461**±0.010 | **0.430**±0.004 |
| | Avg | **0.440**±0.005 | **0.428**±0.001 | **0.372**±0.004 | **0.394**±0.002 | **0.388**±0.001 | **0.385**±0.002 |
| iTransformer | 96 | 0.383±0.005 | 0.396±0.003 | 0.294±0.001 | 0.342±**0.000** | 0.324±**0.001** | 0.351±**0.001** |
| | 192 | 0.437±**0.003** | 0.428±0.002 | 0.376±0.006 | 0.392±0.004 | 0.374±**0.002** | 0.376±**0.002** |
| | 336 | 0.480±0.011 | 0.450±0.009 | 0.421±**0.005** | 0.427±**0.002** | 0.411±**0.001** | 0.401±**0.000** |
| | 720 | 0.493±**0.015** | 0.477±0.009 | 0.423±**0.005** | 0.439±0.002 | 0.480±**0.002** | 0.441±**0.001** |
| | Avg | 0.448±**0.002** | 0.438±0.001 | 0.379±**0.003** | 0.400±0.002 | 0.397±0.001 | 0.393±**0.001** |

## D. Visualization

### D.1. Selection Matrix Visualization

In this section, we provide more visualizations of the SFS module to fully validate its ability of temporal feature extraction.

**Different Prediction Length** As illustrated in Figure 10, the SFS module effectively captures the key frequency components of the spike series. As illustrated in Figure 10, the frequency characteristics of the input sequence with different prediction length are consistent: the 0 frequency component is typically discarded after selection, while frequencies that are multiples of 4 play a crucial role in predicting the final target. These observations align with the findings of FilterNet (Yi et al., 2024), an ANN-based frequency filter model, confirming that our spiking selection mechanism successfully extracts important frequencies from the Fourier spectrum. Notably, the selection rate gradually decreases as the prediction length increases from 96 to 720 timesteps, as longer prediction horizons exhibit weaker correlations with the look-back 96 timesteps,

*Table 10.* 95% confidence intervals of SpikF and iTransformer on ETTm2, Weather and Exchange datasets with MAE loss function. Better performance is highlighted in **bold**.

| Model | | ETTm2 | | Weather | | Exchange | |
|---|---|---|---|---|---|---|---|
| Metric | | MSE | MAE | MSE | MAE | MSE | MAE |
| SpikF | 96 | **0.175±0.001** | **0.251±0.001** | **0.163±0.002** | **0.200±0.001** | **0.084±0.001** | **0.201±0.001** |
| | 192 | **0.242±0.002** | **0.296±0.001** | **0.209±0.002** | **0.241±0.002** | 0.180±0.001 | **0.300±0.001** |
| | 336 | **0.302±0.002** | **0.336±0.002** | **0.266±0.002** | **0.283±0.001** | **0.334±0.020** | **0.417±0.013** |
| | 720 | **0.405±0.007** | **0.397±0.005** | **0.344±0.002** | **0.334±0.002** | **0.841±0.018** | **0.690±0.008** |
| | Avg | **0.281±0.002** | **0.320±0.002** | **0.245±0.000** | **0.265±0.000** | **0.360±0.002** | **0.402±0.001** |
| iTransformer | 96 | 0.182±0.017 | 0.265±0.024 | 0.176±0.004 | 0.216±0.006 | 0.088±0.003 | 0.208±0.003 |
| | 192 | 0.248±0.011 | 0.308±0.019 | 0.224±0.004 | 0.257±0.003 | **0.179**±0.003 | 0.303±0.003 |
| | 336 | 0.313±0.015 | 0.348±0.021 | 0.281±0.002 | 0.299±0.003 | 0.337±**0.003** | 0.421±**0.002** |
| | 720 | 0.411±0.009 | 0.403±0.010 | 0.359±0.003 | 0.350±0.002 | 0.851±0.025 | 0.698±0.013 |
| | Avg | 0.289±0.012 | 0.331±0.018 | 0.260±0.002 | 0.280±0.002 | 0.364±0.008 | 0.407±0.005 |

thereby diminishing the influence of the original spike trains.

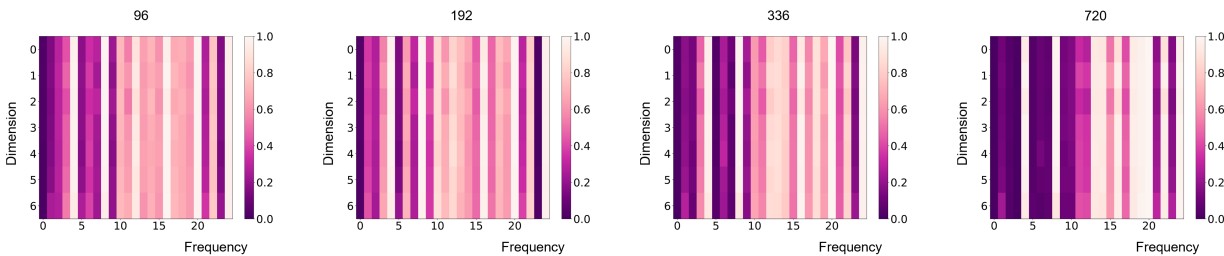

*Figure 10.* Visualization of the Spiking Selector mechanism across varying prediction horizons. The heatmap illustrates selection probabilities. The analysis is conducted on the ETTh1 test set with a fixed look-back window of 96 timesteps and multiple prediction horizons (96, 192, 336, and 720 timesteps). The horizontal axis represents frequency components, while the vertical axis denotes the multidimensional features of the dataset.

**Shared and Individual Spiking Selector Generator**   As illustrated in Figure 11, the individual spiking generator extracts key frequency components for each dimension of the ETTh1 dataset. In comparison, the common generator strategy successfully identifies and highlights the shared significant frequency components across all dimensions, as the components selected by the common generator tend to overlap with the highlighted components of each individual dimension. By balancing the contributions of these components, the shared generator strategy achieves superior performance compared to the individual generator. This analysis is further supported by the superior performance through weight sharing strategy of previous works (Xu et al., 2024; Yi et al., 2024).

### D.2. Visualization of Prediction Results of Different Models

To provide an intuitional comparison among different models, we present visualized prediction results on ETTh1 datasets in Figure 12. The evaluated models include iTransformer (Liu et al., 2024), DLinear (Zeng et al., 2023), and our proposed SpikF. The results clearly demonstrate that SpikF achieves significantly more accurate prediction compared to iTransformer and DLinear.

### D.3. Visualization of Encoded Spikes

This section presents a visualization of the encoded spikes across multiple datasets. As depicted in Figure 13, the spike trains generated by the SPE module exhibit a strong alignment with the input sequence in terms of density. This alignment underscores the interpretability of the spike trains and highlights their effectiveness in capturing the essential features of the input sequence. Furthermore, the sparsity of the spike trains suggests that the energy consumption for subsequent processing

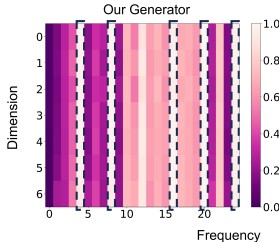 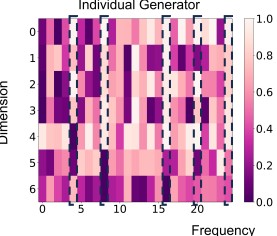

*Figure 11.* Comparison of shared and individual generator strategies. The frequency components selected by the shared generator are framed by dashed lines, along with the corresponding MSE and MAE provided at the bottom of the figure. The look-back window and prediction horizon are set as 96.

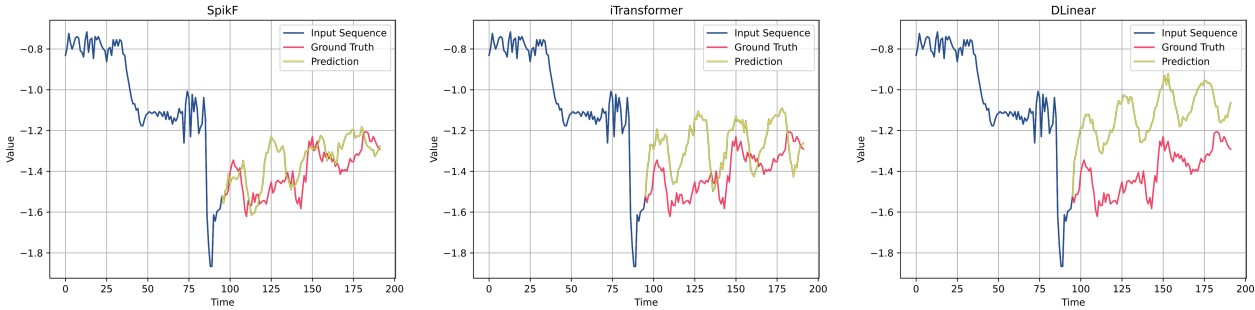

*Figure 12.* Visualization of prediction results on the ETTh1 dataset, comparing SpikF with other benchmark models under a look-back window of 96 and a prediction horizon of 96.

will be minimal, rendering the approach highly efficient.

For these datasets, the prediction horizon is set to 96. The input sequence has been preprocessed through instance normalization (Kim et al., 2021). To enhance clarity, the averaged spike trains have been resized to match the mean and variance of the absolute input sequence. This preprocessing step ensures a more intuitional comparison of the trends between averaged spike trains and input sequences.

## E. Full Results

### E.1. Long-term Prediction

This section presents a comprehensive evaluation of long-term forecasting performance across eight benchmark datasets: ECL, Weather, ETT (ETTh1, ETTh2, ETTm1, ETTm2), Traffic, and Exchange. The detailed performance metrics are organized in Table 11, which offers a comparative analysis of the predictive capabilities of various models. Notably, on ETTh1 dataset, SpikF achieves the lowest MSE and MAE across all prediction lengths.

The results of all benchmarks except SpikF are extracted from (Liu et al., 2024).

### E.2. Extended Benchmark Analysis on ETT Datasets

Given SpikF's remarkable performance on ETT datasets, we expand our evaluation by incorporating additional FFT-based ANN and SNN-based benchmarks to comprehensively assess SpikF's forecasting capabilities.

For FFT-based ANN models, we select FEDformer (Zhou et al., 2022), which employs a frequency domain self-attention mechanism to enhance feature extraction, and FITS (Xu et al., 2024), which utilizes a frequency domain complex linear layer for input sequence upsampling. These models represent advanced approaches in frequency domain analysis. On the SNN side, we include Spiking Recurrent Neural Network (SpikeRNN) (Kim et al., 2019) and Inverted Spiking Transformer (iSpikformer) (Lv et al., 2024b) as benchmarks, as their ANN counterparts are widely recognized for their effectiveness in

long-term prediction tasks. SNN-based models are equipped with SPE. Results of FITS are derived from (Yi et al., 2024) and results of FEDformer are sourced from (Liu et al., 2024).

The results, presented in Table 12, demonstrate that SpikF achieves a precision improvement of $1.4\%$ and $0.7\%$ in terms of MSE and MAE respectively, compared to the second-best SNN-based model. Furthermore, MSE and MAE of SpikF are reduced by $3.4\%$ and $3.6\%$, relative to the second-best FFT-based model.

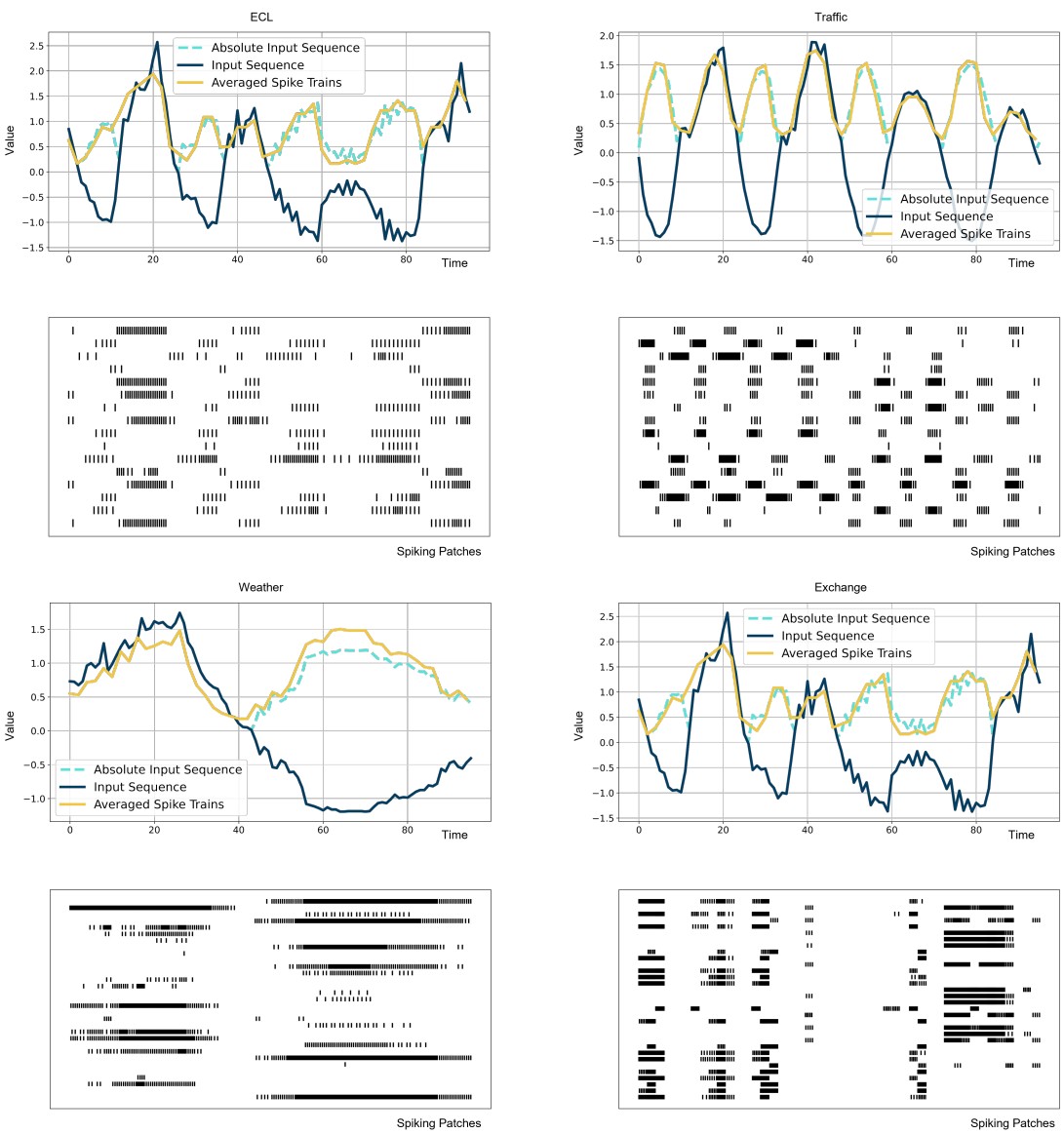

*Figure 13.* Visualization of encoded spike trains alongside the input sequence for the ECL, Traffic, Weather and Exchange datasets. The sparse yet interpretable representation of the spike trains underscores the module's capability to extract meaningful features while maintaining low energy consumption.

*Table 11.* Full results of long-term prediction.

| Model | | SpikF | | iTransformer | | RLinear | | PatchTST | | Crossformer | | TimesNet | | DLinear | | SCINet | | Autoformer | |
|---|---|---|---|---|---|---|---|---|---|---|---|---|---|---|---|---|---|---|---|
| Metric | | MSE | MAE | MSE | MAE | MSE | MAE | MSE | MAE | MSE | MAE | MSE | MAE | MSE | MAE | MSE | MAE | MSE | MAE |
| ECL | 96 | 0.156 | 0.252 | **0.148** | **0.240** | 0.201 | 0.281 | 0.181 | 0.270 | 0.219 | 0.314 | 0.168 | 0.272 | 0.197 | 0.282 | 0.247 | 0.345 | 0.201 | 0.317 |
| | 192 | 0.169 | 0.262 | **0.162** | **0.253** | 0.201 | 0.283 | 0.188 | 0.274 | 0.231 | 0.322 | 0.184 | 0.289 | 0.196 | 0.285 | 0.257 | 0.355 | 0.222 | 0.334 |
| | 336 | 0.188 | 0.281 | **0.178** | **0.269** | 0.215 | 0.298 | 0.204 | 0.293 | 0.246 | 0.337 | 0.198 | 0.300 | 0.209 | 0.301 | 0.269 | 0.369 | 0.231 | 0.338 |
| | 720 | **0.219** | **0.306** | 0.225 | 0.317 | 0.257 | 0.331 | 0.246 | 0.324 | 0.280 | 0.363 | 0.220 | 0.320 | 0.245 | 0.333 | 0.299 | 0.390 | 0.254 | 0.361 |
| | Avg | 0.183 | 0.275 | **0.178** | **0.270** | 0.219 | 0.298 | 0.205 | 0.290 | 0.244 | 0.334 | 0.192 | 0.295 | 0.212 | 0.300 | 0.268 | 0.365 | 0.227 | 0.338 |
| Weather | 96 | 0.163 | **0.200** | 0.174 | 0.214 | 0.192 | 0.232 | 0.177 | 0.218 | **0.158** | 0.230 | 0.172 | 0.220 | 0.196 | 0.255 | 0.221 | 0.306 | 0.266 | 0.336 |
| | 192 | 0.209 | **0.241** | 0.221 | 0.254 | 0.240 | 0.271 | 0.225 | 0.259 | **0.206** | 0.277 | 0.219 | 0.261 | 0.237 | 0.296 | 0.261 | 0.340 | 0.307 | 0.367 |
| | 336 | **0.266** | **0.283** | 0.278 | 0.296 | 0.292 | 0.307 | 0.278 | 0.297 | 0.272 | 0.335 | 0.280 | 0.306 | 0.283 | 0.335 | 0.309 | 0.378 | 0.359 | 0.395 |
| | 720 | **0.344** | **0.334** | 0.358 | 0.347 | 0.364 | 0.353 | 0.354 | 0.348 | 0.398 | 0.418 | 0.365 | 0.359 | 0.345 | 0.381 | 0.377 | 0.427 | 0.419 | 0.428 |
| | Avg | **0.245** | **0.265** | 0.258 | 0.278 | 0.272 | 0.291 | 0.259 | 0.281 | 0.259 | 0.315 | 0.259 | 0.287 | 0.265 | 0.317 | 0.292 | 0.363 | 0.338 | 0.382 |
| ETTh1 | 96 | **0.379** | **0.391** | 0.386 | 0.405 | 0.386 | 0.395 | 0.414 | 0.419 | 0.423 | 0.448 | 0.384 | 0.402 | 0.386 | 0.400 | 0.654 | 0.599 | 0.449 | 0.459 |
| | 192 | **0.432** | **0.421** | 0.441 | 0.436 | 0.437 | 0.424 | 0.460 | 0.445 | 0.471 | 0.474 | 0.436 | 0.429 | 0.437 | 0.432 | 0.719 | 0.631 | 0.500 | 0.482 |
| | 336 | **0.473** | **0.441** | 0.487 | 0.458 | 0.479 | 0.446 | 0.501 | 0.466 | 0.570 | 0.546 | 0.491 | 0.469 | 0.481 | 0.459 | 0.778 | 0.659 | 0.521 | 0.496 |
| | 720 | **0.474** | **0.459** | 0.503 | 0.491 | 0.481 | 0.470 | 0.500 | 0.488 | 0.653 | 0.621 | 0.521 | 0.500 | 0.519 | 0.516 | 0.836 | 0.699 | 0.514 | 0.512 |
| | Avg | **0.440** | **0.428** | 0.454 | 0.447 | 0.446 | 0.434 | 0.469 | 0.454 | 0.529 | 0.522 | 0.458 | 0.450 | 0.456 | 0.452 | 0.747 | 0.647 | 0.496 | 0.487 |
| ETTh2 | 96 | 0.290 | **0.336** | 0.297 | 0.349 | **0.288** | 0.338 | 0.302 | 0.348 | 0.745 | 0.584 | 0.340 | 0.374 | 0.333 | 0.387 | 0.707 | 0.621 | 0.346 | 0.388 |
| | 192 | **0.367** | **0.385** | 0.380 | 0.400 | 0.374 | 0.390 | 0.388 | 0.400 | 0.877 | 0.656 | 0.402 | 0.414 | 0.477 | 0.476 | 0.860 | 0.689 | 0.456 | 0.452 |
| | 336 | **0.414** | **0.420** | 0.428 | 0.432 | 0.415 | 0.426 | 0.426 | 0.433 | 1.043 | 0.731 | 0.452 | 0.452 | 0.594 | 0.541 | 1.000 | 0.744 | 0.482 | 0.486 |
| | 720 | **0.416** | **0.436** | 0.427 | 0.445 | 0.420 | 0.440 | 0.431 | 0.446 | 1.104 | 0.763 | 0.462 | 0.468 | 0.831 | 0.657 | 1.249 | 0.838 | 0.515 | 0.511 |
| | Avg | **0.372** | **0.394** | 0.383 | 0.407 | 0.374 | 0.398 | 0.387 | 0.407 | 0.942 | 0.684 | 0.414 | 0.427 | 0.559 | 0.515 | 0.954 | 0.723 | 0.450 | 0.459 |
| ETTm1 | 96 | **0.317** | **0.345** | 0.334 | 0.368 | 0.355 | 0.376 | 0.329 | 0.367 | 0.404 | 0.426 | 0.338 | 0.375 | 0.345 | 0.372 | 0.418 | 0.438 | 0.505 | 0.475 |
| | 192 | 0.372 | **0.372** | 0.377 | 0.391 | 0.391 | 0.392 | **0.367** | 0.385 | 0.450 | 0.451 | 0.374 | 0.387 | 0.380 | 0.389 | 0.439 | 0.450 | 0.553 | 0.496 |
| | 336 | 0.401 | **0.394** | 0.426 | 0.420 | 0.424 | 0.415 | **0.399** | 0.410 | 0.532 | 0.515 | 0.410 | 0.411 | 0.413 | 0.413 | 0.490 | 0.485 | 0.621 | 0.537 |
| | 720 | 0.461 | **0.430** | 0.491 | 0.459 | 0.487 | 0.450 | **0.454** | 0.439 | 0.666 | 0.589 | 0.478 | 0.450 | 0.474 | 0.453 | 0.595 | 0.550 | 0.671 | 0.561 |
| | Avg | 0.388 | **0.385** | 0.407 | 0.410 | 0.414 | 0.407 | **0.387** | 0.400 | 0.513 | 0.496 | 0.400 | 0.406 | 0.403 | 0.407 | 0.485 | 0.481 | 0.588 | 0.517 |
| ETTm2 | 96 | **0.175** | **0.251** | 0.180 | 0.264 | 0.182 | 0.265 | **0.175** | 0.259 | 0.287 | 0.366 | 0.187 | 0.267 | 0.193 | 0.292 | 0.286 | 0.274 | 0.255 | 0.339 |
| | 192 | 0.242 | **0.296** | 0.250 | 0.309 | 0.246 | 0.304 | **0.241** | 0.302 | 0.414 | 0.492 | 0.249 | 0.309 | 0.284 | 0.362 | 0.399 | 0.445 | 0.281 | 0.340 |
| | 336 | **0.302** | **0.336** | 0.311 | 0.348 | 0.307 | 0.342 | 0.305 | 0.343 | 0.597 | 0.542 | 0.321 | 0.351 | 0.369 | 0.427 | 0.637 | 0.591 | 0.339 | 0.372 |
| | 720 | 0.405 | **0.397** | 0.412 | 0.407 | 0.407 | 0.398 | **0.402** | 0.400 | 1.730 | 1.042 | 0.408 | 0.403 | 0.554 | 0.522 | 0.960 | 0.735 | 0.433 | 0.432 |
| | Avg | **0.281** | **0.320** | 0.288 | 0.332 | 0.286 | 0.327 | **0.281** | 0.326 | 0.757 | 0.610 | 0.291 | 0.333 | 0.350 | 0.401 | 0.571 | 0.537 | 0.327 | 0.371 |
| Traffic | 96 | 0.477 | 0.286 | **0.395** | **0.268** | 0.649 | 0.389 | 0.462 | 0.295 | 0.522 | 0.290 | 0.593 | 0.321 | 0.650 | 0.396 | 0.788 | 0.499 | 0.613 | 0.388 |
| | 192 | 0.481 | 0.289 | **0.417** | **0.276** | 0.601 | 0.366 | 0.466 | 0.296 | 0.530 | 0.293 | 0.617 | 0.336 | 0.598 | 0.370 | 0.789 | 0.505 | 0.616 | 0.382 |
| | 336 | 0.499 | 0.295 | **0.433** | **0.283** | 0.609 | 0.369 | 0.482 | 0.304 | 0.558 | 0.305 | 0.629 | 0.336 | 0.605 | 0.373 | 0.797 | 0.508 | 0.622 | 0.337 |
| | 720 | 0.533 | 0.312 | **0.467** | **0.302** | 0.647 | 0.387 | 0.514 | 0.322 | 0.589 | 0.328 | 0.640 | 0.350 | 0.645 | 0.394 | 0.841 | 0.523 | 0.660 | 0.408 |
| | Avg | 0.497 | 0.296 | **0.428** | **0.282** | 0.626 | 0.378 | 0.481 | 0.304 | 0.550 | 0.304 | 0.620 | 0.336 | 0.625 | 0.383 | 0.804 | 0.509 | 0.628 | 0.379 |
| Exchange | 96 | **0.084** | **0.201** | 0.086 | 0.206 | 0.093 | 0.217 | 0.088 | 0.205 | 0.256 | 0.367 | 0.107 | 0.234 | 0.088 | 0.218 | 0.267 | 0.396 | 0.197 | 0.323 |
| | 192 | 0.180 | 0.300 | 0.177 | **0.299** | 0.184 | 0.307 | **0.176** | **0.299** | 0.470 | 0.509 | 0.226 | 0.344 | 0.176 | 0.315 | 0.351 | 0.459 | 0.300 | 0.369 |
| | 336 | 0.334 | 0.417 | 0.331 | 0.417 | 0.351 | 0.432 | **0.301** | **0.397** | 1.268 | 0.883 | 0.367 | 0.448 | 0.313 | 0.427 | 1.324 | 0.853 | 0.509 | 0.524 |
| | 720 | 0.841 | **0.690** | 0.847 | 0.691 | 0.886 | 0.714 | 0.901 | 0.714 | 1.767 | 1.068 | 0.964 | 0.746 | **0.839** | 0.695 | 1.058 | 0.797 | 1.447 | 0.941 |
| | Avg | 0.360 | **0.402** | 0.360 | 0.403 | 0.378 | 0.417 | 0.367 | 0.404 | 0.940 | 0.707 | 0.416 | 0.443 | **0.354** | 0.414 | 0.750 | 0.626 | 0.613 | 0.539 |

Table 12. More benchmarks on ETT dataset.

| Model | | SpikF | | SpikeRNN | | iSpikformer | | FITS | | FEDformer | |
|---|---|---|---|---|---|---|---|---|---|---|---|
| Metric | | MSE | MAE | MSE | MAE | MSE | MAE | MSE | MAE | MSE | MAE |
| ETTh1 | 96 | 0.379 | 0.391 | 0.392 | 0.395 | 0.410 | 0.408 | 0.386 | 0.396 | 0.376 | 0.419 |
| | 192 | 0.432 | 0.421 | 0.437 | 0.424 | 0.459 | 0.438 | 0.436 | 0.423 | 0.420 | 0.448 |
| | 336 | 0.473 | 0.441 | 0.482 | 0.447 | 0.514 | 0.461 | 0.478 | 0.444 | 0.459 | 0.465 |
| | 720 | 0.474 | 0.459 | 0.499 | 0.471 | 0.511 | 0.476 | 0.502 | 0.495 | 0.506 | 0.507 |
| | Avg | 0.440 | 0.428 | 0.452 | 0.434 | 0.473 | 0.446 | 0.451 | 0.440 | 0.440 | 0.460 |
| ETTh2 | 96 | 0.290 | 0.336 | 0.295 | 0.335 | 0.308 | 0.349 | 0.295 | 0.350 | 0.358 | 0.397 |
| | 192 | 0.367 | 0.385 | 0.375 | 0.387 | 0.382 | 0.397 | 0.381 | 0.396 | 0.429 | 0.439 |
| | 336 | 0.414 | 0.420 | 0.422 | 0.422 | 0.439 | 0.433 | 0.426 | 0.438 | 0.496 | 0.487 |
| | 720 | 0.416 | 0.436 | 0.428 | 0.438 | 0.438 | 0.439 | 0.431 | 0.446 | 0.463 | 0.474 |
| | Avg | 0.372 | 0.394 | 0.380 | 0.396 | 0.392 | 0.407 | 0.383 | 0.408 | 0.437 | 0.449 |
| ETTm1 | 96 | 0.317 | 0.345 | 0.326 | 0.351 | 0.337 | 0.361 | 0.355 | 0.375 | 0.379 | 0.419 |
| | 192 | 0.372 | 0.372 | 0.376 | 0.377 | 0.380 | 0.383 | 0.392 | 0.393 | 0.426 | 0.441 |
| | 336 | 0.401 | 0.394 | 0.396 | 0.394 | 0.415 | 0.406 | 0.424 | 0.414 | 0.445 | 0.459 |
| | 720 | 0.461 | 0.430 | 0.459 | 0.433 | 0.476 | 0.443 | 0.487 | 0.449 | 0.543 | 0.490 |
| | Avg | 0.388 | 0.385 | 0.389 | 0.389 | 0.402 | 0.398 | 0.415 | 0.408 | 0.448 | 0.452 |
| ETTm2 | 96 | 0.175 | 0.251 | 0.176 | 0.251 | 0.178 | 0.256 | 0.183 | 0.266 | 0.203 | 0.287 |
| | 192 | 0.242 | 0.296 | 0.243 | 0.296 | 0.243 | 0.298 | 0.247 | 0.305 | 0.269 | 0.328 |
| | 336 | 0.302 | 0.336 | 0.302 | 0.335 | 0.304 | 0.338 | 0.307 | 0.342 | 0.325 | 0.366 |
| | 720 | 0.405 | 0.397 | 0.409 | 0.396 | 0.406 | 0.398 | 0.407 | 0.399 | 0.421 | 0.415 |
| | Avg | 0.281 | 0.320 | 0.283 | 0.319 | 0.283 | 0.322 | 0.286 | 0.328 | 0.305 | 0.349 |

