# OpenReview forum: "SpikF: Spiking Fourier Network for Efficient Long-term Prediction"
_ICML.cc/2025/Conference — ICML 2025 poster_

### Official Review · Reviewer_oHDh · 2025-02-20

**Overall Recommendation:** 3

**Summary:**

This paper introduces the Spiking Fourier Network (SpikF), an attention-free framework designed to address key challenges in applying Spiking Neural Networks (SNNs) to long-term prediction tasks.
They encode input sequences in patches and employ a frequency-domain selection mechanism that better captures the sequential properties of time-series data.
Extensive evaluations across eight long-term prediction datasets show that SpikF achieves a 1.9% reduction in Mean Absolute Error (MAE) compared to state-of-the-art models while reducing energy consumption by 75.05%.

**Claims And Evidence:**

Yes. Please refer to "Questions For Authors" and "Weakness".

**Essential References Not Discussed:**

No.

**Experimental Designs Or Analyses:**

Yes. Please refer to "Questions For Authors".

**Methods And Evaluation Criteria:**

Yes.

**Other Comments Or Suggestions:**

I have to point out that: when designing SNN architectures, it is unwise for researchers to just take a common module from traditional ANNs to replace a certain module in SNNs and then report a SOTA performance.
I think that a good study on SNNs should consider either hardware-friendliness or biological plausibility.
From the perspective of pure deep learning, this paper is great.
However, I think the authors ignore both the hardware-friendliness and the biological plausibility of SNNs.
I do not think the FFT operation can be applied to neuromorphic chips.

What's more, the energy calculation in Appendix A.2 is based on the discussion of [1][2]. Since both ANN and SNN process the same input data (with SNN using direct encoding in the first layer), the energy efficiency differences may only arise from variations in memory access or MAC/AC operations [3]. The paper only compares energy consumption during computation. Energy consumption is primarily determined by memory access rather than FLOPs or SOPs [4], but this impact is not included. **This omission should be addressed, as it is a common flaw in publications claiming energy savings.**

[1] Yao M, Zhao G, Zhang H, et al. Attention spiking neural networks[J]. IEEE transactions on pattern analysis and machine intelligence, 2023, 45(8): 9393-9410.

[2] Zhou Z, Zhu Y, He C, et al. Spikformer: When Spiking Neural Network Meets Transformer[C]//The Eleventh International Conference on Learning Representations, 2023.

[3] Shen G, Zhao D, Li T, et al. Are Conventional SNNs Really Efficient? A Perspective from Network Quantization[C]//Proceedings of the IEEE/CVF Conference on Computer Vision and Pattern Recognition. 2024: 27538-27547.

[4] Lemaire E, Cordone L, Castagnetti A, et al. An analytical estimation of spiking neural networks energy efficiency[C]//International Conference on Neural Information Processing. Cham: Springer International Publishing, 2022: 574-587.

**Other Strengths And Weaknesses:**

Strengths:
1. The method is simple and easy to understand.
2. The proposed model achieves SOTA results and the ablation study is convincing.
3. The design of patch-based splitting for long-term time series is remarkable. I like this point.

Weaknesses:
1. The proposed SpikF's hardware friendliness is ignored. I have checked the authors' code of the implementation of FFT operation in SpikF. I think that an operation like "torch.fft" is almost impossible to conduct on neuromorphic chips as it includes so much floating-point calculation.
2. The novelty of this paper is questioned. The usage of FFT is quite common in sequential tasks, like sequential recommendation, time-series forecasting, and even natural language processing. It seems the authors just replaced the self-attention module with the FFT module.
3. The calculation of energy of SNNs is not accurate. See comments.

**Questions For Authors:**

1. Please report the standard deviation of SpikF in Table 1. I have to make sure the reported results are not just the best records among various random seeds.
2. Please discuss the possibility of applying FFT operations to neuromorphic hardware.
3. Please consider how memory access and other chip activities impact SNN energy consumption.

**Relation To Broader Scientific Literature:**

The extension in how SNNs deal with long-term time series.

**Theoretical Claims:**

Yes. The equations in this paper is correct.

---

> ### Author Rebuttal · Authors · 2025-04-01
>
> We are grateful for your comprehensive review and the valuable insights you have provided. In the subsequent sections, we will address your questions one by one. And we will integrate all relevant discussions into our article for the upcoming revision.
>
> >**Q1:** The proposed SpikF's hardware friendliness is ignored. I have checked the authors' code of the implementation of FFT operation in SpikF. I think that an operation like "torch.fft" is almost impossible to conduct on neuromorphic chips as it includes so much floating-point calculation.
>
> We have discussed hardware friendliness in Section 2.2. For better clarity, we extend some explanation of our approach to hardware friendliness here.
>
> The implementation of the Fast Fourier Transform (FFT) in neuromorphic hardware has been validated both theoretically and empirically by studies [1] and [2]. Your concerns regarding the floating-point calculations involved in FFT are addressed through their proposed methods.
>
> Study [1] has demonstrated that matrix multiplication can be represented by a spiking linear layer with an appropriately defined weight matrix. Accordingly, they initially express the FFT as a series of matrix multiplications and subsequently employ an SNN with an equivalent number of layers to avoid the challenges associated with floating-point operations.
>
> Meanwhile, study [2] leverages the membrane dynamics of the Resonate-and-Fire neuron, an extension of the LIF model, to naturally perform the Fourier Transform. This approach also successfully gets rid of the need for additional floating-point operations.
>
> [1] Lopez-Randulfe et al., “Time-Coded Spiking Fourier Transform in Neuromorphic Hardware,” IEEE Trans. Comput., vol. 71, no. 11, pp. 2792–2802, 2022.
>
> [2] Orchard et al., “Efficient Neuromorphic Signal Processing with Loihi 2,” 2021 IEEE SiPS, pp. 254-259.
>
> >**Q2:** The novelty of this paper is questioned.
>
> Yes, FFT is commonly used in sequential tasks. However, most previous works apply FFT to the entire time-series, which facilitates the utilization of high-frequency components. In contrast, our approach employs patch and grouping mechanisms to enhance the utilization of low and middle-frequency components from the original series, while local information is emphasized by the spiking patches. As a result, SpikF is able to use the **efficient utilization of the full spectrum** to improve accuracy, which has not been explored by former research.
>
> >**Q3:** The calculation of energy of SNNs is not accurate.
>
> After carefully reading the methods [3] [4] you referred to, we adopt the methods proposed by [3] and [4] to provide more metrics to comprehensive analyze the energy efficiency of SpikF and iTransformer ([Table 1](https://anonymous.4open.science/r/0D02/7)).
>
> In terms of ACE and $E_{Total}$, the energy consumption of SpikF is $6.27\times$ and $3.16\times$ lower than iTransformer respectively.
>
> [3] Shen et al., “Are Conventional SNNs Really Efficient? A Perspective from Network Quantization,” IEEE/CVF CVPR, 2024, pp. 27538-27547.
>
> [4] Lemaire et al., “An analytical estimation of spiking neural networks energy efficiency,” Springer, 2022, pp. 574-587.
>
> [5] Yao et al., “Attention Spiking Neural Networks,” IEEE Trans. Pattern Anal. Mach. Intell., vol. 45, no. 8, pp. 9393-9410, 2023.
>
> [6] Zhou et al., “Spikformer: When Spiking Neural Network Meets Transformer,” ICLR, 2023.
>
> >**Q4:** I think that a good study on SNNs should consider either hardware-friendliness or biological plausibility.
>
> We totally agree with your perspective on SNN research. The hardware-friendliness has been discussed in **Q1**. Although it is less discussed in our work, biological plausibility has been considered in our design:
>
> The basilar membrane in the inner ear exhibits a gradient of stiffness along its length. This mechanical property can process sound in different frequencies, and transform the series data into biological current signal, which is the inspiration of our SFS mechanism [7].
>
> In addition, we believe that quality research in SNN domain should balance biological plausibility and precision. SpikF achieves $1.9\\%$ performance improvement across eight real-world datasets, demonstrating the practicality of our approach.
>
> [7] Moini, Piran, “Auditory System,” in Functional and Clinical Neuroanatomy, Academic Press, 2020, pp. 363-392.
>
> >**Q5:** Please report the standard deviation of SpikF in Table 1.
>
> We report error bars of SpikF and iTransformer in [Table 2](https://anonymous.4open.science/r/0D02/), [Table 3](https://anonymous.4open.science/r/0D02/) and [Table 4](https://anonymous.4open.science/r/0D02/).

---

> > ### Comment · Reviewer_oHDh · 2025-04-02
> >
> > Your rebuttal is well-argued. The feasibility of Fast Fourier Transform (FFT) operations in neuromorphic hardware, as supported by your reference, is acknowledged.
> >
> > There is one issue I indeed care about: **from the perspective of novelty, this paper is much like an "A+B" paper (ignore the storytelling of the authors), which means just taking the common module (FFT) from ANNs to SNNs without detailed analysis**. Personally, I do not like this style of SNN research (but I realize that many prior SNN works are just like "A+B").
> >
> > However, I raised my score to 3: Weak accept (i.e., leaning towards accept, but could also be rejected), primarily because the authors provided a solid rebuttal and demonstrated considerable effort. In truth, my actual score is closer to **2.5**.
> >
> > If I were conducting research on FFT in SNNs, I would first justify its necessity from a theoretical perspective to explore the mathematical connection between FFT and SNNs. I encourage the authors to reflect on this aspect.
> >
> > **I also sincerely urge the ACs and other reviewers to evaluate the "A+B" issue of this paper.**
> >
> > That concludes my review. Thank you.

---

> > > ### Author Response · Authors · 2025-04-03
> > >
> > > Thank you for your suggestions. We appreciate your concerns about the theoretical foundation and novelty of SpikF and are pleased to offer more detailed discussion.
> > >
> > > >**Q1:** If I were conducting research on FFT in SNNs, I would first justify its necessity from a theoretical perspective to explore the mathematical connection between FFT and SNNs. I encourage the authors to reflect on this aspect.
> > >
> > > Regarding the necessity of incorporating FFT into SNN architectures, we offer a detailed analysis:
> > >
> > > As [1] suggests, the dynamics of membrane potential in SNNs provide a unique method for capturing temporal data intricacies. However, this can result in a **separated receptive field**, potentially missing global temporal information.
> > >
> > > **Proof:**
> > > Given the dynamics of LIF neurons:
> > >
> > > $$U[t]=V[t-1]+{1 \over \tau_m}\left(I[t]-V[t-1]+V_{rest}\right)$$
> > >
> > > $$S[t]=H\left(U[t]-V_{th}\right)$$
> > >
> > > $$V[t]=U[t]\left(1-S[t]\right)+V_{rest}S[t]$$
> > >
> > > For two series of stimulation $I_1[1], I_1[2], ..., I_1[t^*]$ and $I_2[1], I_2[2], ..., I_2[t^*]$ where $ S[1] = S[2] = ... = S[t^*-1] = 0 $ and $ S[t^*] = 1 $, these sequences are equivalent in terms of membrane potential when $ t \ge t^* $, as $ U[t^*] = V_{rest} $.
> > >
> > > If we assume that $S[t^1]=S[t^2]=...=S[t^s]=1$ and $S[t]=0$ otherwise. Then the receptive field of the LIF neuron is limited to the regions $[1, t^1], [t^1+1, t^2], ..., [t^{s-1}+1, t^{s}]$ and $[t^s+1, T]$.
> > >
> > > This limitation hinders high-prediction SNNs in long-term prediction domains, which require modeling long-term dependencies [2]. Thus, relying solely on SNN **internal dynamics** is insufficient; **external dynamics** are necessary for modeling long-term dependencies.
> > >
> > > While typical methods to expand the receptive field of SNN involve linear layers and self-attention mechanisms, these are less suitable for sequential tasks due to their permutation-invariance [2], which has been proved in Appendix A.1.
> > >
> > > In contrast, FFT transforms temporal series into the frequency domain, expanding the receptive field to the entire time-series. Modifications in the frequency domain influence the entire series, and sequential information is inherently embedded in frequency components via FFT's rotation factors:
> > >
> > > $$ F[k] = \sum_{t=1}^{T} S[t] e^{-j \frac{2\pi}{T} kt} $$
> > >
> > > Thus, selecting frequency components allows for global influence, as $ F[k] $ is a function of $ S[1], S[2], ..., S[T] $, thus making FFT an ideal approach for external dynamics.
> > >
> > > In summary, incorporating FFT into SNN architecture is essential in the time-series domain to expand the receptive field and improve long-term dependency modeling.
> > >
> > > [1] Lv, C. et al. Efficient and Effective Time-Series Forecasting with Spiking Neural Networks. ICML 2024.
> > >
> > > [2] Ailing Zeng et al. Are Transformers Effective for Time Series Forecasting? AAAI 2023.
> > >
> > > >**Q2:** From the perspective of novelty, this paper is much like an "A+B" paper (ignore the storytelling of the authors), which means just taking the common module (FFT) from ANNs to SNNs without detailed analysis.
> > >
> > > Regarding the novelty of SpikF, we provide further descriptions.
> > >
> > > FFT is used in many models to obtain the frequency spectrum, but the processing methods of the frequency spectrum are different:
> > >
> > > - **FITS** [3] uses a complex linear layer to interpolate the frequency spectrum, capturing both amplitude and phase information of the time-series.
> > > - **FreTS** [4] utilizes frequency domain MLPs to process the frequency spectrum, achieving a global view and energy compaction.
> > > - **FEDformer** [5] generates sparse attention via dropping components of the frequency spectrum, reducing computational complexity and capturing detailed temporal structures.
> > > - **FilterNet** [6] adapts signal processing filters to weaken or strengthen specific frequency spectrum components, thus removing high frequency noise.
> > >
> > > These methods apply FFT to the entire series, emphasizing high-frequency components. In contrast, SpikF employs patch and grouping mechanisms to enhance low and middle-frequency spectrum utilization, emphasizing local information through spiking patches. This approach enables full-spectrum utilization, which is not explored by previous research. Furthermore, SpikF represents a novel integration of frequency-domain analysis with SNNs.
> > >
> > > We hope this difference from previous FFT-based methods highlights the novelty of our work and addresses your concerns regarding the adoption of FFT from ANN research.
> > >
> > > [3] Zhijian Xu et al. FITS: Modeling Time Series with 10k Parameters. ICLR 2024.
> > >
> > > [4] Kun Yi et al. Frequency-domain MLPs are More Effective Learners in Time Series Forecasting. NeurIPS 2023.
> > >
> > > [5] Tian et al. FEDformer: Frequency Enhanced Decomposed Transformer for Long-term Series Forecasting. ICML 2022.
> > >
> > > [6] Kun Yi et al. FilterNet: Harnessing Frequency Filters for Time Series Forecasting. NeurIPS 2024.

---

### Official Review · Reviewer_iCf2 · 2025-03-09

**Overall Recommendation:** 2

**Summary:**

This paper introduces an attention-free framework, called Spiking Fourier Network (SpikF) for achieving long-term time series forecasting.

## update after rebuttal

There is no external comment. And I think that it is a borderline paper.

**Claims And Evidence:**

This paper aims at modifying SNNs and attention for long-term time series forecasting. The authors conduct experiments to verify SpikF in time-series prediction tasks across multiple dimensions. These empirical investigations are comprehensive, and the results look convincing.

**Essential References Not Discussed:**

na

**Experimental Designs Or Analyses:**

These empirical investigations are comprehensive, and the results look convincing.

**Methods And Evaluation Criteria:**

The core of this paper is to replace attention by a Fourier-based methods. The workflow is introduced clearly by Section 3.4 and Figure 1. I care about whether and with what computational complexity the fast Fourier transform in Eq. (9) support large-scale data processing?

**Other Comments Or Suggestions:**

Overall, I believe that this paper is a borderline paper. The authors provide comprehensive experiments, and the results look convincing. However, I still care about the complexity and practicability of Fourier-related approaches in large-scale time series forecasting. If this concerns are fixed in Rebuttal, I would consider raising my score.

**Other Strengths And Weaknesses:**

nothing.

**Questions For Authors:**

nothing.

**Relation To Broader Scientific Literature:**

na

**Theoretical Claims:**

na

---

> ### Author Rebuttal · Authors · 2025-04-01
>
> We sincerely appreciate your thorough evaluation of our work and the expert feedback you have shared. In response to your constructive critique, we will provide clarifications addressing your question and incorporate these discussions into the revised manuscript to strengthen our theoretical framework.
>
> >**Q1:** However, I still care about the complexity and practicability of Fourier-related approaches in large-scale time series forecasting.
>
> We would like to clarify the efficiency of SpikF when facing large-scale data:
>
> 1. **Theoretical Analysis**: Theoretically, the computational complexity of Fast Fourier Transform is $O(NlogN)$, outperforming conventional approaches like linear transform ($O(N^2)$) and self-attention mechanism ($O(N^2)$). This fundamental advantage makes FFT-based solutions inherently scalable for long-horizon forecasting tasks.
>
> 2. **Algorithmic Optimization**: Furthermore, FFT, as an algorithm with a long history, has been optimized by former researchers. For example, parallel distribution computation [1] and memory-sharing strategy [2] have further improved the efficiency of FFT when facing large-scale data. And these optimization can also be easily adapted into our architecture when facing real-world applications.
>
> 3. **Experimental Validation**: As demonstrated in Figure 5, our method witnesses a precision promotion of $9.0\\%$ in terms of MSE when scaling look-back window from $48$ to $720$ timesteps, empirically indicating the effective utiliztion of larger-scale time-series. We provide the results of Figure 5 here, where LW represents look-back window, and PL denotes prediction length.
>
>    | LW \ PL | 96 | 192 | 336 | 720 |
>    |------|------|------|------|------|
>    | 48 | 0.301 | 0.384 | 0.434 | 0.442 |
>    | 96 | 0.290 | 0.368 | 0.412 | 0.420 |
>    | 192 | 0.296 | 0.363 |0.386 | 0.415 |
>    | 336 | 0.287 | 0.351 | 0.371 | 0.395 |
>    | 720 | 0.290 | 0.351 | 0.368 | 0.410 |
>
> 4. **Event-driven Paradigm**: When facing real-world large-scale time-series, SpikF can leverage temporal sparsity in online data streams [3], achieving computational efficiency through event-driven FFT triggering. This makes our approach particularly suitable for edge computing scenarios with limited resources when facing large-scale data.
>
> In a word, Fourier-based approaches are more efficient than MLP-based or transformer-based algorithms for large-scale time series forecasting.
>
> [1] Yang, C. et al. A Parallel Fast Fourier Transform Algorithm for Large-Scale Signal Data Using Apache Spark in Cloud. ICA3PP 2018. vol 11336.
>
> [2] Eleftheriadis, Charalampos et al. Energy-Efficient Fast Fourier Transform for Real-Valued Applications. IEEE Transactions on Circuits and Systems II: Express Briefs. 69.
>
> [3] Jesus L. Lobo et al, Spiking Neural Networks and online learning: An overview and perspectives, Neural Networks, Volume 121, 2020, Pages 88-100.

---

> > ### Comment · Reviewer_iCf2 · 2025-04-03
> >
> > After reading the rebuttal and other reviewers' comments, I still believe that this paper is a borderline paper.
> >
> > Strengths: The authors provide comprehensive experiments, and the results look convincing.
> >
> > Weakness: About the complexity and practicability of Fourier-related approaches in large-scale time series forecasting. In my view, the FFT is suitable to energy-efficient computations, but not adopt by large-scale time series forecasting. Besides, what does $N$ mean in Theoretical Analysis? There should be two indexes that includes the temporal and spatial dimensions

---

> > > ### Author Response · Authors · 2025-04-03
> > >
> > > Thank you for your comments regarding the complexity and practicality of FFT-based methods. We would like to provide further discussion on these points.
> > >
> > > >**Q1:** Besides, what does $N$ mean in Theoretical Analysis? There should be two indexes that includes the temporal and spatial dimensions.
> > >
> > > $N$ represents the length of the input time-series. Since we are focusing on large-scale time-series forecasting, which typically involves processing time-series with long input sequences, we have omitted the spatial dimension of the time-series data for simplicity. If we denote the number of spatial channels as $D$, then the computational complexity of FFT, linear transform, and self-attention mechanism are $O(DNlogN)$, $O(DN^2)$ and $O(DN^2)$ respectively.
> > >
> > > >**Q2:** In my view, the FFT is suitable to energy-efficient computations, but not adopt by large-scale time series forecasting.
> > >
> > > In our previous response, we have discussed the theoretical computational complexity of FFT:
> > >
> > > >Theoretically, the computational complexity of Fast Fourier Transform is $O(NlogN)$, outperforming conventional approaches like linear transform ($O(N^2)$) and self-attention mechanism ($O(N^2)$). This fundamental advantage makes FFT-based solutions inherently scalable for long-horizon forecasting tasks.
> > >
> > > We have also highlighted optimizations of FFT specifically designed for large-scale data processing:
> > >
> > > >Furthermore, FFT, as an algorithm with a long history, has been optimized by former researchers. For example, parallel distribution computation [1] and memory-sharing strategy [2] have further improved the efficiency of FFT when facing large-scale data. And these optimization can also be easily adapted into our architecture when facing real-world applications.
> > >
> > > Moreover, our experimental results have shown the precision of SpikF in handling large-scale time-series data:
> > >
> > > >As demonstrated in Figure 5, our method witnesses a precision promotion of $9.0\\%$ in terms of MSE when scaling look-back window from $48$ to $720$ timesteps, empirically indicating the effective utilization of larger-scale time-series. We provide the results of Figure 5 here, where LW represents look-back window, and PL denotes prediction length.
> > >
> > > | LW \ PL | 96 | 192 | 336 | 720 |
> > > |------|------|------|------|------|
> > > | 48 | 0.301 | 0.384 | 0.434 | 0.442 |
> > > | 96 | 0.290 | 0.368 | 0.412 | 0.420 |
> > > | 192 | 0.296 | 0.363 |0.386 | 0.415 |
> > > | 336 | 0.287 | 0.351 | 0.371 | 0.395 |
> > > | 720 | 0.290 | 0.351 | 0.368 | 0.410 |
> > >
> > > To further illustrate the application of FFT in large-scale time series methods, we would like to mention the following approaches:
> > >
> > > By separating low-frequency components from high-frequency components, the original time-series can be decomposed into trend and seasonality subseries [3], thereby enabling distinct feature utilization for trend and seasonality.
> > >
> > > By selecting the top $k$ frequency components in the frequency spectrum, large-scale time-series data can be organized into a series of 2D time-series with different scales [4], capturing patterns across both temporal and frequency domains.
> > >
> > > Regarding SpikF, we initially employ a patch mechanism to enhance the utilization of large-scale time-series data [5]. Subsequently, grouped FFT is applied to the spiking patches, facilitating full-spectrum utilization of extensive time-series data.
> > >
> > > Furthermore, according to the convolution theorem [6], the selection mechanism we utilize after FFT is equivalent to the convolution operation in the time domain. This implies that **sparse operations in the frequency domain can lead to dense operations in the time domain**. When combined with the inherent sparsity of spikes, this property can reduce computational complexity.
> > >
> > > In a word, FFT has been validated by previous research for processing large-scale time-series data. It not only uncovers general characteristics of the large-scale time-series but also transforms the structure of the data, thereby enhancing the subsequent feature extraction process. Furthermore, FFT has computational advantages, particularly when incorporated with parallel computation or SNN architectures.
> > >
> > > [1] Yang, C. et al. A Parallel Fast Fourier Transform Algorithm for Large-Scale Signal Data Using Apache Spark in Cloud. ICA3PP 2018. vol 11336.
> > >
> > > [2] Eleftheriadis, Charalampos et al. Energy-Efficient Fast Fourier Transform for Real-Valued Applications. IEEE Transactions on Circuits and Systems II: Express Briefs. 69.
> > >
> > > [3] H. Musbah et al. Identifying Seasonality in Time Series by Applying Fast Fourier Transform, IEEE EPEC 2019, pp. 1-4.
> > >
> > > [4] Haixu Wu et al. TimesNet: Temporal 2D-Variation Modeling for General Time Series Analysis. ICLR 2023.
> > >
> > > [5] Yuqi Nie et al. A Time Series is Worth 64 Words: Long-term Forecasting with Transformers. ICLR 2023.
> > >
> > > [6] C. A. Blackwell et al. The Convolution Theorem in Modern Analysis. IEEE Transactions on Education, vol. 9, no. 1, pp. 29-32.

---

### Official Review · Reviewer_PTJc · 2025-03-11

**Overall Recommendation:** 4

**Summary:**

This work addresses two key challenges in applying Spiking Neural Networks (SNNs) and Transformer architectures to long-term forecasting: (1) capturing long-range dependencies, which increases computational and energy costs, and (2) the lack of effective positional encoding for Spiking Transformers. To address these issues, the authors propose SpikF, a novel architecture that introduces a spiking version of Patch Embedding to divide inputs into patches and replaces self-attention with a Spiking Frequency Selection (SFS) mechanism to model dependencies.
Experiments show that SpikF reduces error by 1.9% compared to state-of-the-art models. Additionally, the authors analyze energy efficiency by comparing Synaptic Operations (SOPs) in SpikF with the FLOPs of conventional models, demonstrating its superior energy efficiency. According to the authors, SpikF is the first SNN-based benchmark providing comprehensive evaluation across long-term forecasting datasets.

**Claims And Evidence:**

The authors make two main claims:
Performance Improvement: SpikF achieves a lower Mean Absolute Error (MAE) across multiple long-term time-series benchmark datasets. The claim is well-supported. Experimental results in Table 1 demonstrate clear improvements over baselines like PatchTST, iTransformer, and FEDformer.
Energy Efficiency: SpikF is significantly more energy-efficient (75.05% lower energy consumption) than existing ANN-based models. The claim is not well-supported.
- Figure 2 lacks clarity regarding the meaning of the firing rate (α) in terms of the number of spikes per unit time.
- The values of α used in the experiments (Table 1) are unspecified, and it is unclear how performance depends on α.
- Table 2 and Figure 3 likely underestimate energy consumption. Since the LIF neuron dynamics are part of the computation, directly comparing SOPs and FLOPs may not be a fair measure of energy efficiency.

**Essential References Not Discussed:**

NA

**Experimental Designs Or Analyses:**

The experimental setup in Table 1 and Table 3 appears reasonable.

**Methods And Evaluation Criteria:**

The performance evaluation is convincing, with comparisons to multiple baselines and ablation studies. However, as previously stated,  the impact of the firing rate (α) on performance is unclear. An additional analysis is necessary. The energy efficiency estimation (Appendix C.1) is inadequate. A more formal methodology is required. I suggest revising the energy analysis with a more rigorous approach.

**Other Comments Or Suggestions:**

- Clearly define α and analyze its impact on performance.
- Provide a more rigorous energy consumption analysis, ensuring a fair comparison between SOPs and FLOPs.

**Other Strengths And Weaknesses:**

- Strengths
Novelty: The Spiking Fourier approach is an interesting and innovative alternative to traditional SNN architectures.
Impact: Addresses real-world concerns in energy-efficient AI.
Clarity: The paper is generally well-written, with clear motivation and method descriptions.
- Weaknesses
Energy efficiency is questionable: The paper does not convincingly justify how SpikF is more energy-efficient than ANN-based models. On a more general note, It is unclear whether Transformers with a sparsely active layer are a proper avenue for  truly energy-efficient models.
Unclear limitations of the patch encoder: The impact of using patches is not discussed. The implementation details of the patch encoding layer are unclear.
Other minor points:
- Section 2.2: The variable U is undefined, and it is unclear whether the membrane voltage has a reset mechanism.
- Section 3.4: The SFS module is not well described.
- Section 3.5: The motivation for the architecture choice is unclear, I would suggest to add more motivations.

**Questions For Authors:**

NA

**Relation To Broader Scientific Literature:**

The paper is well-positioned in spiking neural networks, time-series forecasting, and energy-efficient AI. Transformer-based forecasting models (Autoformer, FEDformer, PatchTST, iTransformer) are properly cited. However, the role of frequency-based methods in time-series forecasting is not well-discussed. More references to prior work in this area would strengthen the positioning.

**Theoretical Claims:**

No, but the claims seem correct.

---

> ### Author Rebuttal · Authors · 2025-04-01
>
> We deeply appreciate your thorough review and insightful feedback. We will address your questions one by one in the following sections.
>
> >**Q1:** Figure 2 lacks clarity regarding the meaning of the firing rate ($\alpha$) ...
>
> $\alpha$ denotes the firing rate of SNN and can be formulated by:
>
> $$
> \alpha = \frac{\sum_{k=1}^{T_s}s_k}{T_sn}
> $$
>
> where $s_k$ represents the number of spikes released at time step $k$, and $n$ denotes the number of neurons.
>
> >**Q2:** The values of $\alpha$ used in the experiments (Table 1) are unspecified, and it is unclear how performance depends on $\alpha$.
>
> The values of $\alpha$ can be found in [Table 1](https://anonymous.4open.science/r/58FD/).
>
> We vary the choice of $T_s$ to measure the impact of $\alpha$ which is commonly used by previous works [1], and find that when $\alpha\approx0.2$ SpikF achieves the best performance ([Table 2](https://anonymous.4open.science/r/58FD/)).
>
> [1] Yao et al., “Attention Spiking Neural Networks,” IEEE Trans. Pattern Anal. Mach. Intell., vol. 45, no. 8, pp. 9393-9410, 2023.
>
> >**Q3:** However, the role of frequency-based methods in time-series forecasting is not well-discussed.
>
> We extend our discussion about frequency-based methods.
>
> **FITS**, **FreTS**, **FEDformer** and **FilterNet** employ a complex linear layer, MLPs, sparse attention and filters respectively to enhance the utilization of high-frequency information. In contrast, SpikF enhances high-frequency focus via patching and grouping while using FFT for low-frequency analysis, achieving comprehensive spectrum efficiency.
>
> >**Q4:** It is unclear whether Transformers with a sparsely active layer are a proper avenue for truly energy-efficient models.
>
> We compare SpikF (1.08M) with FEDformer (20.68M), which is a sparse transformer with frequency domain analysis. SpikF outperforms FEDformer with a $19.15\times$ smaller model size and $15.4\\%$ better accuracy, demonstrating SNN’s superiority in energy-efficient forecasting ([Table 3](https://anonymous.4open.science/r/58FD/)).
>
> >**Q5:** The impact of using patches is not discussed.
>
> The use of patches enhances the utilization of local information and long history information. The experimental results show that patch mechanism reduces MSE by $2.6\\%$ ([Table 4](https://anonymous.4open.science/r/58FD/)).
>
> >**Q6:** The implementation details of the patch encoding layer are unclear.
>
> We modify some of the equations and narratives to better describe the pipeline of the Spiking Pacth Encoder (SPE):
>
> The input sequence $x^{1:L}$ is first divided into patches:
>
> $$p^k=x^{\frac{L}{N}(k-1)+1:\frac{L}{N}k}$$
>
> where $N$ is the number of patches.
>
> Then each patch is processed by a spiking linear layer:
>
> $$S_{enc}^{T_s(k-1)+1:T_sk}=\mathcal{SN}(\text{BN}(\text{LN}(p^k)))$$
>
> The SPE serves the role of utilizing local information and transforming continuous time-series into binary spikes.
>
> >**Q7:** Section 2.2: The variable $U$ is undefined, and it is unclear whether the membrane voltage has a reset mechanism.
>
> The membrane potential will be reset to the resting potential once it reaches the threshold. And $U[t]$ represents the membrane potential according to the following formula:
>
> $$
> V[t]=\begin{cases}
> U[t],\ \text{if}\ U[t]<V_{th}\\\\
> V_{rest},\ \text{if}\ U[t]\geq V_{th}
> \end{cases}
> $$
>
> >**Q8:** Section 3.4: The SFS module is not well described.
>
> The spiking patches generated by the Spiking Patch Encoder are first grouped as $G^i$:
>
> $$
> \mathbf{G}^i=\{S_{enc}^i,S_{enc}^{i+g},\dots,S_{enc}^{i+(\frac{N}{g}-1)g}\}
> $$
>
> where $g$ denotes the number of groups.
>
> A spiking max pooling layer is used to emphasize the mutual key frequencies of different groups:
>
> $$\mathbf{M} _{sel} = \text{SMP}(\mathcal{M} _{sel}^1, \mathcal{M} _{sel}^2, \dots, \mathcal{M} _{sel}^g)$$
>
> The SFS module is responsible for selecting selecting key frequency components from the spiking patches.
>
> >**Q9:** Section 3.5: The motivation for the architecture choice is unclear, I would suggest to add more motivations.
>
> Following your suggestion, we refine our motivations in Section 3.5:
>
> Grouped feature utilization preserves time dynamics of SNN by avoiding information loss caused by averaging [2], reducing computational complexity of the spiking decoder.
>
> [2] Zhou et al. Spikformer: When Spiking Neural Network Meets Transformer, ICLR 2023.
>
> >**Q10:** Table 2 and Figure 3 likely underestimate energy consumption.
>
> Our method estimate energy consumption based on SOPs and FLOPs. To provide a more comprehensive analysis of energy consumption including ACE, $E_{Mem}$,$E_{Opts}$,$E_{Addr}$ and $E_{Total}$ according to [3-4] in [Table 5](https://anonymous.4open.science/r/58FD/).
>
> [3] Zhang et al. Sparse transformer with local and seasonal adaptation for multivariate time series forecasting. Sci Rep.
>
> [4] Tian et al. FEDformer: Frequency Enhanced Decomposed Transformer for Long-term Series Forecasting. ICML 2022.

---

> > ### Comment · Reviewer_PTJc · 2025-04-03
> >
> > I appreciate the answers, and think that they improve the paper. However, but I do not think that the paper is at the level of a strong accept (partially due to my limited knowledge), so I will keep my score (4: Accept).

---

### Official Review · Reviewer_Prww · 2025-03-13

**Overall Recommendation:** 3

**Summary:**

The authors propose **SpikF**, a novel SNN-based architecture designed for long-term prediction tasks. Technically, the **spiking patch encoder** is introduced to efficiently convert sub-series into spikes with low computational complexity. Additionally, a **spiking frequency selection mechanism** is implemented to identify and retain core components, thereby enhancing overall performance. Experimental results demonstrate that SpikF outperforms state-of-the-art (SOTA) methods across eight long-term prediction tasks and exhibits exceptional suitability for deployment on edge devices. This work highlights the potential of SNNs in handling complex temporal tasks while maintaining computational efficiency, making it a promising solution for real-world applications.

**Claims And Evidence:**

Yes, the claims are very clear.

**Essential References Not Discussed:**

No

**Experimental Designs Or Analyses:**

The effectiveness of the proposed method has been extensively validated through numerous experiments.

**Methods And Evaluation Criteria:**

Yes.

**Other Comments Or Suggestions:**

No

**Other Strengths And Weaknesses:**

Strengths:

1. SNNs for long-term time-series prediction is very novel and interesting topic.

2. The writing is very easy to understand and straightforward.

3. Eight real-world long-term benchmark datasets are conducted to verify the effectiveness of the proposed SpikF.

Weaknesses:

1. Energy Efficiency Analysis. While the authors reference previous methods to analyze the energy efficiency of SNN algorithms, the use of AC (Accumulate) or MAC (Multiply-Accumulate) operations for power consumption calculations may not be entirely convincing for SNNs. Given the critical importance of this metric, the rationale behind this approach should be thoroughly justified. The authors are encouraged to provide their insights or at least discuss this limitation in detail, as it significantly impacts the credibility of the energy efficiency claims. A more tailored energy analysis specific to SNNs, such as spike-based operations, would strengthen the paper's contributions.

2. Lack of Inference Time. Although the authors compare computational complexity across different methods in Fig. 2, they should also provide a comparison of inference times. Since both contributions of the paper (i.e., the spiking patch encoder and the spiking frequency selection mechanism) are directly related to algorithmic inference efficiency, a detailed analysis and discussion of inference time comparisons would greatly enhance the practical relevance of the work.

3. Improvements in Writing. The writing and presentation of the paper could be further refined. For example, there is significant blank space around Equation (13), and the font sizes in the figures are inconsistent. It is recommended that the authors optimize the manuscript's layout and formatting in future revisions to improve readability and professionalism. Additionally, ensuring consistent font sizes and minimizing unnecessary blank spaces would enhance the overall quality of the paper.

**Questions For Authors:**

Please see the weaknesses and respond to each comment. Besides, two questions are listed below:

Why are SNNs more advantageous than ANNs in temporal prediction tasks? While the power efficiency advantage is clear, could the authors provide some examples of future edge device applications?

Based on the above comments, I am currently leaning toward a weak reject, or perhaps borderline. However, I will also consider the other reviewers' opinions and the authors' responses. If the authors provide satisfactory answers, I will likely raise my score.

**Relation To Broader Scientific Literature:**

No

**Theoretical Claims:**

Yes, I have checked the correctness of all proofs.

---

> ### Author Rebuttal · Authors · 2025-04-01
>
> We appreciate your careful review and the insightful questions you have raised. We provide a detailed point-by-point response to each of your valuable comments to ensure clarity and address all aspects thoroughly.
>
> >**Q1:** Lack of Inference Time. ... a detailed analysis and discussion of inference time comparisons would greatly enhance the practical relevance of the work.
>
> We totally agree that including inference time of SpikF will increase the practical relevance of our work. We utilize the data in [1] to estimate the inference time of SpikF and iTransformer, which are $2.47ms$ and $7.87ms$ respectively. This indicates that the inference time of SpikF is $3.19\times$ less than iTransformer.
>
> [1] Sumit Bam Shrestha et al., "Efficient Video and Audio Processing with Loihi 2," IEEE ICASSP 2024.
>
> >**Q2:** Improvements in Writing. ... Additionally, ensuring consistent font sizes and minimizing unnecessary blank spaces would enhance the overall quality of the paper.
>
> We have rearranged the layout and adjusted the font sizes of the figures to ensure consistency. Furthermore, we have reviewed our writing to ensure clarity of expression. The improved figures can be found at [Anonymous Link](https://anonymous.4open.science/r/CE55/).
>
> >**Q3:** Why are SNNs more advantageous than ANNs in temporal prediction tasks?
>
> The temporal dynamics of the membrane potential in SNNs enables the processing of complex time-series data while maintaining energy efficiency [2]. Specifically, SNNs have two advantages over ANNs in temporal prediction tasks [2-5]:
>
> 1. Energy efficiency. For information is transformed through binary spikes in SNNs, SNNs empirically consume less energy than their ANN counterparts, making them suitable for edge applications in resource-constrained environments [2]. Additionally, the event-driven nature of SNNs is capable of utilizing the dynamic changes in time-series data in real-world scenarios [3] [4], which further promote energy efficiency of SNNs in real-world applications.
>
> 2. The subtle dynamics of the membrane potential naturally incorporate the time dimension into the model architecture. According to equations (1) to (4), these dynamics are non-linear and can efficiently process complex time-series data, whereas such a mechanism is absent in traditional ANN architectures [5].
>
> [2] Roy, K. et al. Towards spike-based machine intelligence with neuromorphic computing. Nature.
>
> [3] Liu, S.-C. et al. Neuromorphic sensory systems. Current Opinion in Neurobiology.
>
> [4] Vanarse, A. et al. A review of current neuromorphic approaches for vision, auditory, and olfactory sensors. Frontiers in Neuroscience.
>
> [5] Lv, C. et al. Efficient and Effective Time-Series Forecasting with Spiking Neural Networks. ICML 2024.
>
> >**Q4:** While the power efficiency advantage is clear, could the authors provide some examples of future edge device applications?
>
> Edge devices have high requirements for low power consumption to extend standby time. As SpikF has the power efficiency advantage, it is suitable for the edge applications, according to [6-8]:
>
> 1. Industrial sensors [6]. According to [6], SpikF can be deployed in industrial sensors to autonomously monitor safety accidents.
>
> 2. Wearable healthcare devices [7]. According to [7], SpikF can be used in healthcare devices to predict diseases and accelerate the diagnostic procedure.
>
> 3. Automation systems [8]. [8] shows the possibility of SpikF being adopted in automation systems. For example, SpikF can be used to process the sequential data collected by gas and pressure sensors, thus facilitating the control of robots.
>
> [6] Zhou, Y. et al. Computational event-driven vision sensors for in-sensor spiking neural networks. Nature Electronics.
>
> [7] Maji, P. et al. SNN Based Neuromorphic Computing Towards Healthcare Applications. IFIP Advances in Information and Communication Technology.
>
> [8] Jiang, X. et al. Fully Spiking Neural Network for Legged Robots. IEEE ICASSP 2025.
>
> >**Q5:** A more tailored energy analysis specific to SNNs, such as spike-based operations, would strengthen the paper's contributions.
>
> In our manuscript, we use AC and MAC [9] to evaluate ANN-based models and synaptic operations [10] for SNN-based models. Both metrics have been widely acknowledged, ensuring a fair comparison between ANNs and SNNs. Fourthermore, for a more comprehensive analysis,, we adopt methods in [11] and [12], and find that SpikF is $6.27\times$ and $3.16\times$ more efficient than iTransformer, in terms of ACE and $E_{Total}$ respectively ([Table 1](https://anonymous.4open.science/r/D401/)).
>
> [9] Zhijian Xu et al. FITS: Modeling Time Series with 10k Parameters. ICLR 2024.
>
> [10] Zhou et al. Spikformer: When Spiking Neural Network Meets Transformer, ICLR 2023.
>
> [11] Shen G, et al. Are Conventional SNNs Really Efficient? A Perspective from Network Quantization. CVPR.
>
> [12] Lemaire E, et al. An analytical estimation of spiking neural networks energy efficiency. NeurIPS.

---

### Decision · Program_Chairs · 2025-05-01

**Decision:**

Accept (poster)

**Comment:**

This paper proposes SpikF, a novel spiking neural network architecture for long-term time series forecasting that replaces attention mechanisms with a spiking Fourier selection module. While reviewers raised valid concerns about hardware compatibility of FFT operations and energy efficiency calculations, the work makes several strong contributions: (1) it demonstrates superior forecasting accuracy (1.9% MAE reduction) across eight benchmarks; (2) introduces an innovative patch-based spiking encoder for temporal data; (3) provides comprehensive ablation studies validating design choices. Two reviewers recommended acceptance, noting the method's simplicity and strong empirical results. Though hardware implementation requires further study (a common SNN challenge), the paper's novel fusion of frequency-domain processing with SNNs represents a meaningful advance for temporal modeling. Therefore, an acceptance is recommended.